# WRNIP1 prevents transcription-associated genomic instability

**Pasquale Valenzisi[†], Veronica Marabitti[†], Pietro Pichierri, Annapaola Franchitto***

Section of Mechanisms Biomarkers and Models, Department of Environment and Health, Istituto Superiore di Sanita, Rome, Italy

**Abstract** R-loops are non-canonical DNA structures that form during transcription and play diverse roles in various physiological processes. Disruption of R-loop homeostasis can lead to genomic instability and replication impairment, contributing to several human diseases, including cancer. Although the molecular mechanisms that protect cells against such events are not fully understood, recent research has identified fork protection factors and DNA damage response proteins as regulators of R-loop dynamics. In this study, we identify the Werner helicase-interacting protein 1 (WRNIP1) as a novel factor that counteracts transcription-associated DNA damage upon replication perturbation. Loss of WRNIP1 leads to R-loop accumulation, resulting in collisions between the replisome and transcription machinery. We observe co-localization of WRNIP1 with transcription/replication complexes and R-loops after replication perturbation, suggesting its involvement in resolving transcription-replication conflicts. Moreover, WRNIP1-deficient cells show impaired replication restart from transcription-induced fork stalling. Notably, transcription inhibition and RNase H1 overexpression rescue all the defects caused by loss of WRNIP1. Importantly, our findings highlight the critical role of WRNIP1 ubiquitin-binding zinc finger (UBZ) domain in preventing pathological persistence of R-loops and limiting DNA damage, thereby safeguarding genome integrity.

**\*For correspondence:**
annapaola.franchitto@iss.it

[†]These authors contributed equally to this work

**Competing interest:** The authors declare that no competing interests exist.

## eLife assessment

This **valuable** paper examines the role of the WRNIP1 AAA+ ATPase in regulating R-loop formation, which induces a conflict with active replication forks and transcription. The authors provide **convincing** evidence to support a role of the ubiquitin-binding UBZ domain of WRNIP1 in R-loop suppression generated by this conflict. The work is of interest to researchers who work on genome stability/instability.

## Introduction

The maintenance of genome integrity is essential for cell viability, relying on complete and accurate DNA replication. However, the genome is continuously threatened by several types of DNA lesions that impede replication fork progression, leading to fork stalling, replication stress and ultimately to genomic instability – a hallmark of cancer cells. Different molecular mechanisms underlie replication stress, and understanding their occurrence has been of interest for cancer therapy. Among these mechanisms, the transcription process plays a crucial role in inducing replication fork stalling. Transcription not only hampers replication fork progression but also acts as an enhancer of replication impediments, including protein-DNA complexes, DNA damage, and non-B-form DNA secondary structures (*Gómez-González and Aguilera, 2019*).

R-loops have emerged as critical determinants of transcription-associated replication stress, capable of causing genomic instability and promoting cancer and other human diseases (*Gómez-González*

and Aguilera, 2019; García-Muse and Aguilera, 2019; Crossley et al., 2019). R-loops are three-stranded structures that consist of an DNA/RNA hybrid and a displaced single-stranded DNA. They are involved in various physiological processes, including transcription termination, regulation of gene expression, and DNA repair. Under normal conditions, R-loops are transient structures resolved by enzymes (Sanz et al., 2016; Wahba et al., 2011). However, persistent R-loops can become pathological, leading to the formation of transcription-replication conflicts (TRCs). Given the high frequency of replication and transcription processes in cells, TRCs are particularly likely to occur. The interference between transcription and replication at very large human genes, whose transcription continues into S-phase, may contribute to the instability of common fragile sites (CFS), the most replication stress-sensitive regions in the human genome (Helmrich et al., 2011).

The idea that R-loops cause fork stalling, promoting TRCs and leading to transcription-associated DNA damage is supported by the fact that many strategies, aimed at minimizing collisions and facilitating replication fork progression following unscheduled R-loop accumulation, rely on replication fork protection factors and DNA damage response (DDR) proteins (Shivji et al., 2018; Hatchi et al., 2015; Schwab et al., 2015; García-Rubio et al., 2015; Bhatia et al., 2014; Liang et al., 2019; Urban et al., 2016; Chang et al., 2017; Marabitti et al., 2019; Tresini et al., 2015). However, the precise mechanisms by which cells cope with TRCs resulting from persistent R-loops are still largely unknown.

Human WRN-interacting protein 1 (WRNIP1) is a genome maintenance factor (Yoshimura et al., 2017). WRNIP1 binds to forked DNA that resembles stalled forks (Yoshimura et al., 2009) with its foci overlapping with replication factories (Crosetto et al., 2008), suggesting a role at replication forks. Consistent with this, WRNIP1 protects stalled forks from MRE11-mediated degradation and promotes fork restart upon replication stress (Leuzzi et al., 2016; Porebski et al., 2019). Additionally, WRNIP1 is implicated in ATM-dependent checkpoint activation in response to mild replication stress (Kanu et al., 2016; Marabitti et al., 2020). Notably, in cells with defective ATR-dependent checkpoint activation, such as Werner syndrome cells (Basile et al., 2014), WRNIP1 mediates ATM-dependent CHK1 phosphorylation to counteract pathological R-loop accumulation (Marabitti et al., 2020). Furthermore, WRNIP1 is found to be enriched at CFS, suggesting a role in maintaining their stability (Pladevall-Morera et al., 2019). However, data regarding a role of WRNIP1 in restricting transcription-associated genomic instability are not available.

Here, we reveal that loss of WRNIP1 and the activity of WRNIP1 ubiquitin-binding zinc finger (UBZ) domain are necessary to limit unscheduled R-loops leading to TRCs. Additionally, we establish that loss of WRNIP1 functions is responsible for the increased genomic instability observed in WRNIP1-deficient and UBZ mutant cells upon mild replication perturbation.

## Results

### Transcription-dependent DNA damage occurs in WRNIP1-deficient and UBZ mutant cells upon MRS

To investigate the contribution of WRNIP1 in maintaining genome integrity, we evaluated DNA damage accumulation in response to mild replication stress (MRS) induced by nanomolar dose of the DNA polymerase inhibitor aphidicolin (Aph). In addition to its ATPase activity, which is involved in fork restart (Leuzzi et al., 2016), WRNIP1 also contains a ubiquitin-binding zinc finger (UBZ) domain. Although this domain has been implicated in fork-related functions (Yoshimura et al., 2017), its function is still poorly defined.

In our experiments, we used the SV40-transformed MRC5 fibroblast cell line (MRC5SV), MRC5SV cells stably expressing WRNIP1-targeting shRNA (shWRNIP1), and isogenic cell lines stably expressing the RNAi-resistant full-length wild-type WRNIP1 (shWRNIP1$^{WT}$), its ATPase-dead mutant form (shWRNIP1$^{T294A}$) (Leuzzi et al., 2016), or the UBZ-dead mutant form of WRNIP1 (shWRNIP1$^{D37A}$) that abolishes the ubiquitin-binding activity (Bish and Myers, 2007; Nomura et al., 2012; Figure 1A). Initially, we measured DNA damage by alkaline Comet assay in WRNIP1-deficient and WRNIP1 mutant cells. Consistent with previous experiments (Marabitti et al., 2020), we found that loss of WRNIP1 resulted in higher spontaneous levels of DNA damage compared to wild-type cells (MRC5SV), and that Aph exacerbated this phenotype (Figure 1B). Interestingly, in WRNIP1 mutant cells, spontaneous genomic damage was similar to that observed in WRNIP1-deficient cells but, after MRS, it was significantly enhanced only in UBZ mutant cells (Figure 1B).

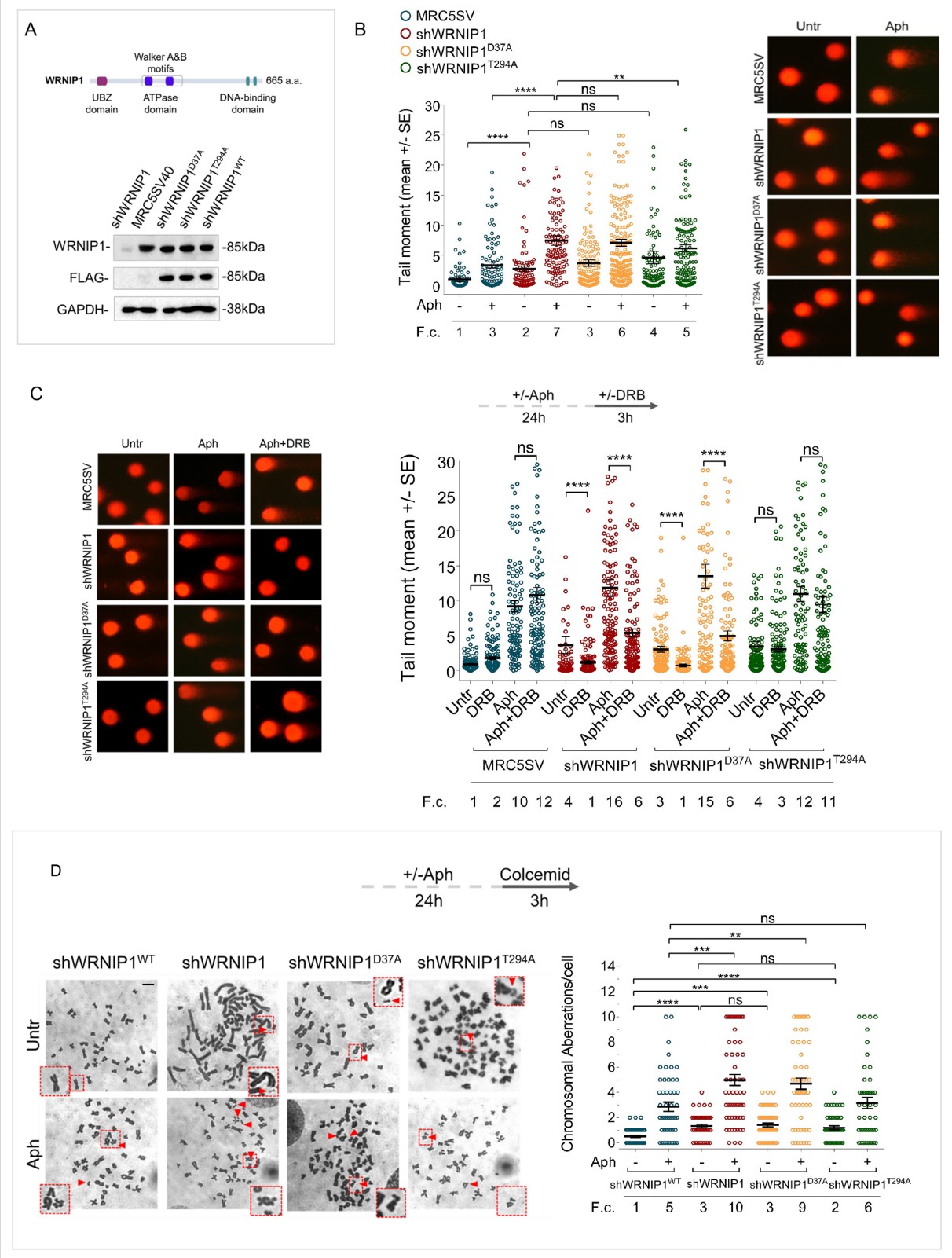

**Figure 1.** Loss of WRNIP1 or its UBZ domain results in DNA damage accumulation and enhanced chromosomal instability upon MRS. (**A**) Schematic representation of human WRNIP1 protein structure. Western blot analysis shows WRNIP1 protein expression in wild-type cells (shWRNIP1[WT]), WRNIP1-deficient cells (shWRNIP1) and WRNIP1 ATPase mutant (shWRNIP1[T294A]) or WRNIP1 UBZ mutant (shWRNIP1[D37A]) cells. MRC5SV40 fibroblasts were used as a positive control. The membrane was probed with an anti-FLAG or anti-WRNIP1 antibody. GAPDH was used as a loading control. (**B**) Analysis of

*Figure 1 continued on next page*

*Figure 1 continued*

DNA damage accumulation evaluated by alkaline Comet assay. MRC5SV, shWRNIP1, shWRNIP1$^{D37A}$, and shWRNIP1$^{T294A}$ cells were treated or not with 0.4 µM Aph for 24 hr, followed by Comet assay. Data are presented as means from three independent experiments. Horizontal black lines represent the mean (ns, not significant; **, p<0.01; ****, p<0.0001; Mann-Whitney test). Representative images are provided. (**C**) Analysis of chromosomal aberrations in the indicated cell lines treated according to the experimental scheme. Dot plot displays the number of chromosomal aberrations per cell ±SE from three independent experiments. Horizontal black lines represent the mean (ns, not significant; **, p<0.01; ***, p<0.001; ****, p<0.0001; two-tailed Student's t test). Representative images, including insets with enlarged metaphases for better visualization of chromosomal aberrations, are included. (**D**) Evaluation of DNA damage accumulation by alkaline Comet assay. Cells were treated according to the experimental scheme and subjected to Comet assay. Dot plot presents as means from three independent experiments. Horizontal black lines represent the mean (ns, not significant; ****, p<0.0001; Mann-Whitney test). Representative images are provided. Scale bar, 2.5 µm. Fold changes (F. c) were calculated as ratio of means between each experimental sample and the relative wild-type untreated.

The online version of this article includes the following source data and figure supplement(s) for figure 1:

**Source data 1.** PDF containing original scans of the western blot (anti-WRNIP1, anti-FLAG, and anti-GAPDH) and related original TIFF images.

**Figure supplement 1.** Loss of WRNIP1 or its UBZ domain increases DNA damage upon MRS.

**Figure supplement 2.** Loss of WRNIP1 or its UBZ domain increases levels of γH2AX.

In parallel experiments, we assessed the presence of chromosomal damage. As shown in *Figure 1C*, shWRNIP1 and WRNIP1 mutant cells exhibited a greater number of chromosomal aberrations under unperturbed conditions compared to their corrected isogenic counterparts (shWRNIP1$^{WT}$). However, while loss of WRNIP1 or mutation of its UBZ domain heightened the average number of gaps and breaks after Aph than their corrected counterparts (shWRNIP1$^{WT}$), that of ATPase activity did not (*Figure 1C*). Therefore, loss of WRNIP1 or its ubiquitin-binding function renders cells highly sensitive to Aph-induced MRS, affecting genomic integrity.

We next wondered whether DNA damage accumulated in a transcription-dependent manner. To test this possibility, we performed a alkaline Comet assay using MRC5SV, shWRNIP1, shWRNIP1$^{D37A}$ and shWRNIP1$^{T294A}$ cells incubated with Aph and/or the 5,6-dichloro-1-ß-D-ribofurosylbenzimidazole (DRB), a strong inhibitor of RNA synthesis, as described (*Marabitti et al., 2019*). Our analysis showed that DNA damage was not sensitive to transcription inhibition at any of the tested conditions in wild-type and WRNIP1 ATPase mutant cells (*Figure 1D*). In contrast, DRB significantly suppressed the amount of DNA damage in both shWRNIP1 and shWRNIP1$^{D37A}$ cells that were either untreated or treated with Aph (*Figure 1D*). Similar results were obtained using another transcription inhibitor, Cordycepin (3'-deoxyadenosine) (*Müller et al., 1977*; *Figure 1—figure supplement 1*). Additionally, anti-phospho-H2AX immunostaining, considered an early sign of DNA damage caused by replication fork stalling (*Ward and Chen, 2001*), confirmed a transcription-dependent DNA damage accumulation in shWRNIP1 and shWRNIP1$^{D37A}$ cells (*Figure 1—figure supplement 2*).

Altogether these findings suggest that WRNIP1 and the activity of its UBZ domain are required to prevent transcription-dependent DNA damage.

## Enhanced R-loop accumulation is observed in WRNIP1-deficient cells

The main structures associated with transcription that can impede fork movement and result in transcription-replication conflicts (TRCs) are R-loops (*Gan et al., 2011*; *Allison and Wang, 2019*; *Gaillard and Aguilera, 2016*). As WRNIP1 may counteract R-loop-mediated TRCs by reducing transcription-dependent DNA damage accumulation, we first assessed R-loops levels in our cells using the well-established anti-RNA-DNA hybrid S9.6 antibody (*Boguslawski et al., 1986*; *Hamperl et al., 2017*; *Marabitti et al., 2019*). We observed a significant increase in nuclear S9.6 intensity in unperturbed WRNIP1-deficient cells, but not in their corrected counterparts (shWRNIP1$^{WT}$; *Figure 2A*). Furthermore, while Aph treatment caused increased R-loop levels in both cell lines, the values were considerably higher in shWRNIP1 cells (*Figure 2A*). Importantly, the S9.6 staining was strongly suppressed upon overexpression of RNase H1 that removes R-loops (*Cerritelli et al., 2003*; *Figure 2A*).

To further confirm the accumulation of R-loop within nuclear DNA, we isolated genomic DNA from shWRNIP1$^{WT}$ and shWRNIP1 cells and performed a dot blot analysis. Consistent with fluorescence analysis, the S9.6 signal was higher in shWRNIP1 cells than in shWRNIP1$^{WT}$ cells and was abolished by RNase H treatment (*Figure 2B*; *Morales et al., 2016*). This result suggest that DNA damage observed in WRNIP1-deficient cells may be correlated with the elevated accumulation of R-loops.

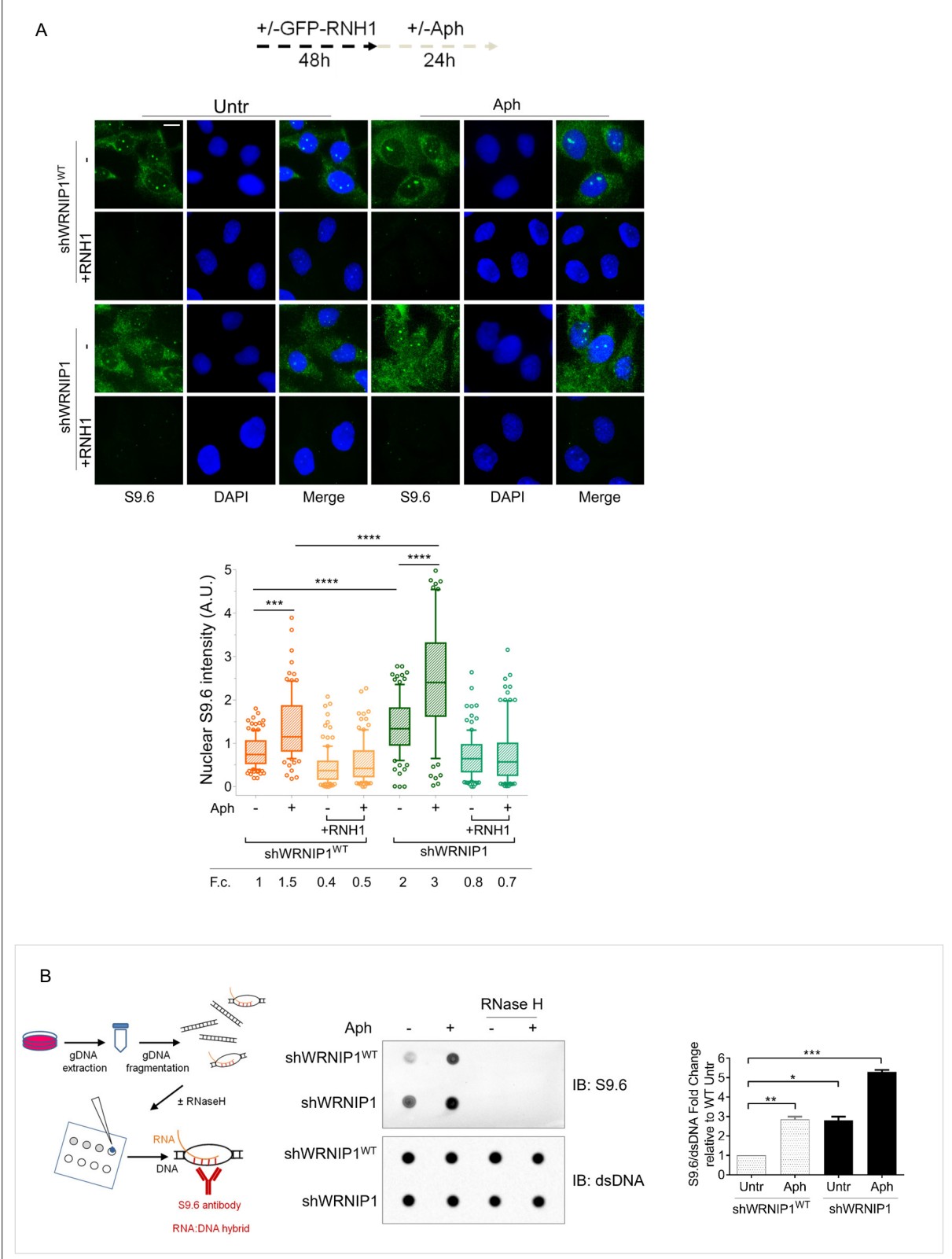

**Figure 2.** Loss of WRNIP1 or its UBZ domain results in R-loop accumulation upon MRS. (**A**) Evaluation of R-loop accumulation by immunofluorescence analysis in shWRNIP1<sup>WT</sup> and shWRNIP1 cells, following treatment as reported in the experimental design post-transfection with GFP-tagged RNase H1 or empty vector. Cells were fixed, stained with anti-RNA-DNA hybrid S9.6 antibody, and DNA counterstained with DAPI. Representative images are provided for each single color channel. Box plot displays nuclear S9.6 fluorescence intensity, with boxes and whiskers representing 20–75 and 10–90

*Figure 2 continued on next page*

Figure 2 continued

percentiles, respectively. The line represents the median value. Data are presented as means from three independent experiments. Horizontal black lines represent the mean. Error bars represent standard error (***, p<0.001; ****, p<0.0001; two-tailed Student's t test). Scale bar, 2.5 μm. (**B**) Dot blot to confirm R-loop accumulation. Genomic DNA isolated from shWRNIP1$^{WT}$ and shWRNIP1 cells, treated as reported in the experimental design, was spotted onto a nitrocellulose membrane. The membranes was probed with anti-RNA-DNA hybrid S9.6 and anti-dsDNA antibodies. Treatment with RNase H was used as a negative control. Representative gel images of at least three replicates are shown (* p<0.05; ** p<0.01; *** p<0.001; two-tailed Student's t test). Fold changes (F. c) were calculated as ratio of means between each experimental sample and the relative wild-type untreated.

The online version of this article includes the following source data for figure 2:

**Source data 1.** PDF containing original scans of the dot blot (anti-S9.6, anti-dsDNA) and related original TIFF images used for the analysis.

## Loss of WRNIP1 or its UBZ domain results in R-loop-dependent DNA damage upon MRS

After demonstrating that WRNIP1-deficient cells accumulate high levels of R-loops, we investigated the risks posed to genome integrity by subjecting them to different replication stress conditions. As observed above, loss of WRNIP1 led to a slight increase in spontaneous DNA damage, while Aph enhanced the comet tail moment to a greater extent in shWRNIP1 than in MRC5SV cells (*Figure 3A*). A similar trend was observed using hydroxyurea (HU) as a replication-perturbing agent (*Figure 3A*). Overexpression of RNase H1 strongly reduced the tail moment, particularly in shWRNIP1 cells, exhibiting greater efficacy following Aph-induced replication slowdown compared to arrest by HU, where the difference is not significant (*Figure 3A*). This confirms R-loops as a driver of genome instability in WRNIP1-deficient cells and suggests that R-loop-mediated DNA damage becomes evident following replication stress that does not completely block replication fork progression.

To remove contaminant-free RNA, especially dsRNA that might interfere with the capture of RNA-DNA hybrids by the S9.6 antibody (*Hartono et al., 2018*), and to enhance the accuracy of R-loop level determination, cells were subjected to RNase III treatment following established protocols (*Crossley et al., 2021*). We initially verified the specificity of the enzyme under our conditions. RNase III treatment substantially decreased the level of dsRNA, nearly eliminating it, as confirmed by assessing specific antibody staining against dsRNA (*Figure 3—figure supplement 1*).

Next, through RNase III treatment, we investigated which of the WRNIP1 activities could prevent the formation of aberrant R-loops upon MRS. We observed similar levels of spontaneous R-loops in cells with mutations in either the ATPase or UBZ domain. However, following MRS, shWRNIP1$^{D37A}$ cells exhibited a more pronounced S9.6 signal than shWRNIP1$^{T294A}$ cells (*Figure 3B*), indicating a role for the UBZ domain of WRNIP1 in counteracting R-loop accumulation during MRS.

Finally, we aimed to determine whether R-loop-mediated DNA damage in WRNIP1-deficient cells was due to loss of the UBZ domain of WRNIP1. Notably, we found that suppression of DNA damage in shWRNIP1$^{D37A}$ cells upon RNase H1 overexpression was similar to that observed in shWRNIP1 cells (*Figure 3C*), suggesting that the ubiquitin-binding activity of WRNIP1 is necessary to prevent R-loop-mediated DNA damage upon MRS.

## The UBZ domain of WRNIP1 is required to attenuate TRCs upon MRS

Since TRCs play a crucial role in promoting R-loop-mediated genomic instability (*García-Muse and Aguilera, 2016*), we investigated the occurrence of such conflicts under our conditions. We employed a proximity ligation assay (PLA), a well-established method for detecting physical interactions (*Söderberg et al., 2008*). Antibodies against proliferating cell nuclear antigen (PCNA) and RNA polymerase II (RNA pol II) were utilized to label replication forks and transcription complexes, respectively, as previously described (*Hamperl et al., 2017*). Our analysis revealed a higher number of spontaneous PLA signals (indicated by red spots) in WRNIP1-deficient and UBZ mutant cells compared to wild-type cells (*Figure 4*). Importantly, although Aph increased the co-localization of PCNA and RNA pol II in all cell lines, the number of PLA spots was significantly greater in shWRNIP1 and shWRNIP1$^{D37A}$ cells than in shWRNIP1$^{WT}$ cells (*Figure 4*). This phenotype was substantially reduced by overexpressing RNase H1 (*Figure 4*), suggesting that the UBZ domain of WRNIP1 may play a role in attenuating R-loop-induced TRCs upon MRS. The unexpected increase in TRCs upon RNase H1 overexpression in Aph-treated wild-type cells may be explained with the disruption of repair proteins, potentially leading to heightened fork stalling and increased conflicts (*Shen et al., 2017*).

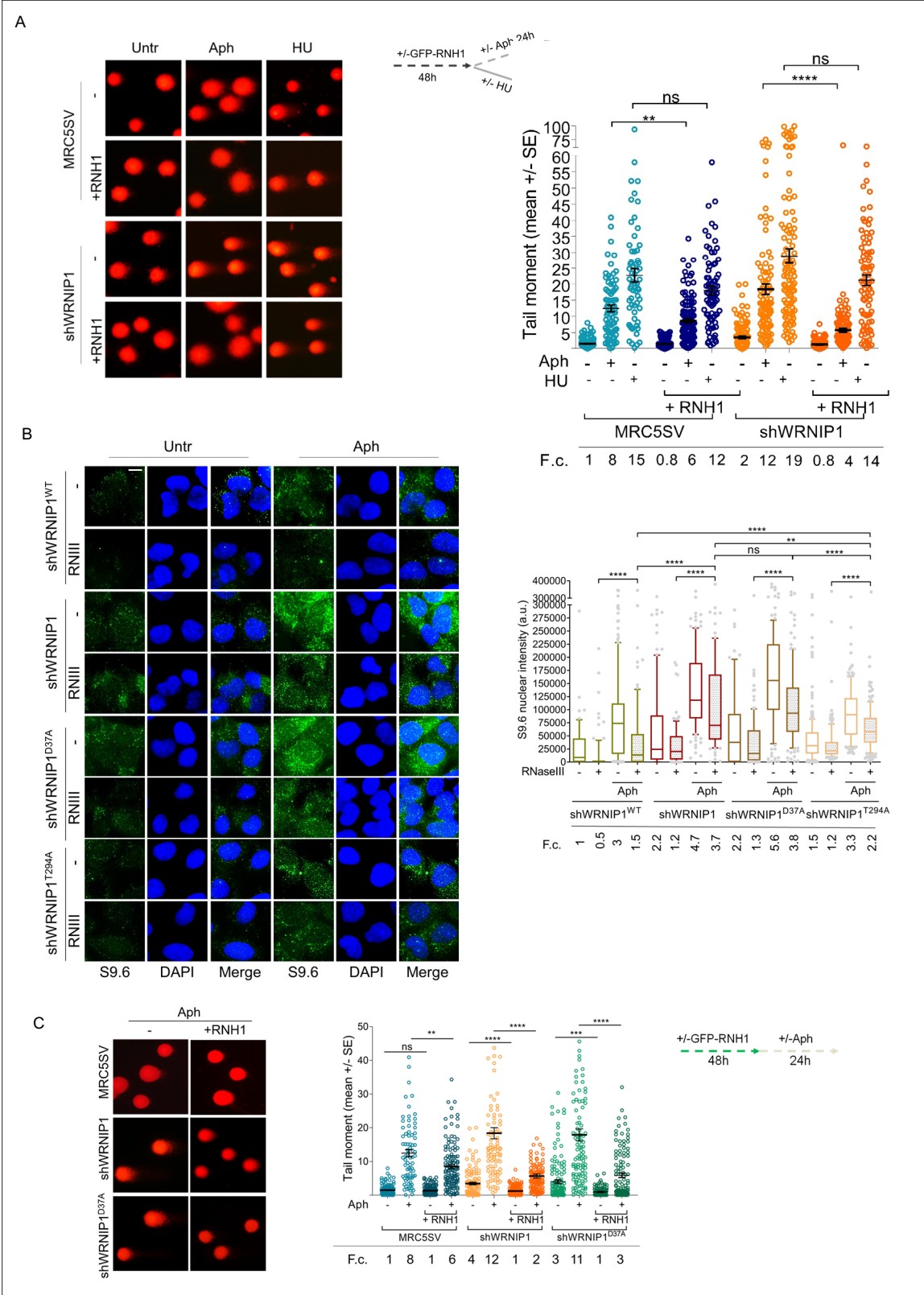

**Figure 3.** Loss of WRNIP1 or its UBZ domain leads to R-loop-dependent accumulation upon MRS. (**A**) Analysis of DNA damage accumulation by alkaline Comet assay. MRC5SV and shWRNIP1 cells, post-transfection with GFP-tagged RNaseH1 or empty vector (-), were treated or not with Aph or HU, following the experimental scheme. Subsequently, they were subjected to Comet assay. Dot plot shows data from three independent experiments (ns, not significant; **, p<0.01; **** p<0.0001; Mann-Whitney test). Representative images are provided. (**B**) Immunofluorescence analysis to determine R-

*Figure 3 continued on next page*

*Figure 3 continued*

loop levels in shWRNIP1[WT], shWRNIP1, shWRNIP1[D37A], and shWRNIP1[T294A] cells, treated or not with 0.4 µM Aph for 24 hr. After fixing, cells were subjected or not to RNase III digestion and stained with anti-RNA-DNA hybrid S9.6 antibody. Representative images are provided for each single color channel. Nuclei were counterstained with DAPI. Box plot displays nuclear S9.6 fluorescence intensity, with boxes and whiskers representing 20–75 and 10–90 percentiles, respectively. The line represents the median value. Horizontal black lines represent the mean. Error bars represent SE (ns, not significant; **, p<0.01; ****, p<0.0001; two-tailed Student's t test). Scale bar, 2.5 µm. (**C**) Analysis of the effect of R-loop resolution on DNA damage accumulation using alkaline Comet assay. Cells, post-transfection with GFP-tagged RNase H1 or empty vector, were treated as reported in the experimental scheme and subjected to Comet assay. Dot plot represents data from three independent experiments. Horizontal black lines represent the mean (ns, not significant; ** p<0.01; *** p<0.001; **** p<0.0001; Mann-Whitney test). Representative images are provided. Fold changes (F.c) were calculated as ratio of means between each experimental sample and the relative wild-type untreated.

The online version of this article includes the following figure supplement(s) for figure 3:

**Figure supplement 1.** RNase III digestion significantly reduces the amount of dsRNA.

Next, we examined the localization of WRNIP1 near/at transcription and replication machineries by conducting PLA assays in shWRNIP1[WT] and shWRNIP1[D37A] cells. For this purpose, antibodies against WRNIP1 and RNA pol II or PCNA, respectively, were used. Our findings support the notion that WRNIP1 is indeed localized at the sites of TRCs following treatment with Aph in wild-type cells, as evidenced by an increased number of PLA spots between WRNIP1 and RNA pol II (*Figure 5A*), as well as between WRNIP1 and PCNA (*Figure 5B*). Surprisingly, a similar, but more pronounced, phenotype was observed in WRNIP1 UBZ mutant cells compared to wild-type cells, both under normal conditions and after treatment (*Figure 5A and B*).

Furthermore, we explored the localization of WRNIP1 in proximity to R-loops. Through the PLA assay, we noticed a growing number of spots, indicating interactions between WRNIP1 and R-loops (anti-S9.6) (*Figure 5C*). Interestingly, this phenomenon was more evident in WRNIP1 UBZ mutant cells (*Figure 5C*), suggesting that WRNIP1, along with its UBZ domain, plays a role in responding to R-loop accumulation.

One of the initial steps in resolving R-loops involves the recruitment of RAD51 to stalled forks (*Chappidi et al., 2020*). Our PLA assay revealed that RAD51-R-loops nuclear spots were clearly observed in both MRC5SV and shWRNIP1[D37A] cell lines following Aph treatment, but their presence was reduced in WRNIP1 UBZ mutant cells compared to wild-type cells (*Figure 5B*).

Taken together, these findings suggest that WRNIP1 co-localizes with transcription/replication complexes and R-loops following MRS. Additionally, they indicate that the ubiquitin-binding function of WRNIP1 helps mitigate R-loop-mediated TRCs.

## Transcription-mediated R-loop formation can act as a barrier to DNA replication in cells lacking WRNIP1 or its UBZ domain activity

R-loops are transcription-associated structures that can impede the progression of replication forks (*Belotserkovskii et al., 2018*). To assess the direct impact of R-loops on fork progression, we performed a DNA fiber assay to examine replication fork dynamics at the single molecule level in Aph-treated MRC5SV, shWRNIP1, and shWRNIP1[D37A] cells. Cells were labeled sequentially with the thymidine analogues 5-chloro-2'-deoxyuridine (CldU) and 5-iodo-2'-deoxyuridine (IdU), as described in the scheme (*Figure 5A*). Under normal growth conditions, and in agreement with previous data (*Leuzzi et al., 2016*), MRC5SV and shWRNIP1 cells exhibited nearly identical fork velocities, while WRNIP1 UBZ mutant cells showed a significantly reduced velocity (*Figure 6A*). Following Aph treatment, fork velocity decreased in all cell lines, but values were markedly lower in cells lacking WRNIP1 or its UBZ domain (*Figure 6B*). Importantly, RNase H1 overexpression led to a significant increase in the rate of fork progression only in shWRNIP1 and shWRNIP1[D37A] cells (*Figure 6A and B*).

DNA fiber analysis also revealed that loss of WRNIP1 or its UBZ domain resulted in a greater percentage of stalled forks induced by Aph compared to control cells (*Figure 6C*). Moreover, when comparing the percentage of restarting forks in all cell lines, we observed that the absence of WRNIP1 reduced the ability of cells to resume replication after release from Aph to the same extent as loss of its UBZ domain (*Figure 6D*). Suppressing R-loop formation lowered the percentage of stalled forks and, consistently, increased that of restarting forks in both shWRNIP1 and shWRNIP1[D37A] cells (*Figure 6C and D*). Similar results were obtained by treating cells with DRB (*Figure 6—figure supplement 1A–D*).

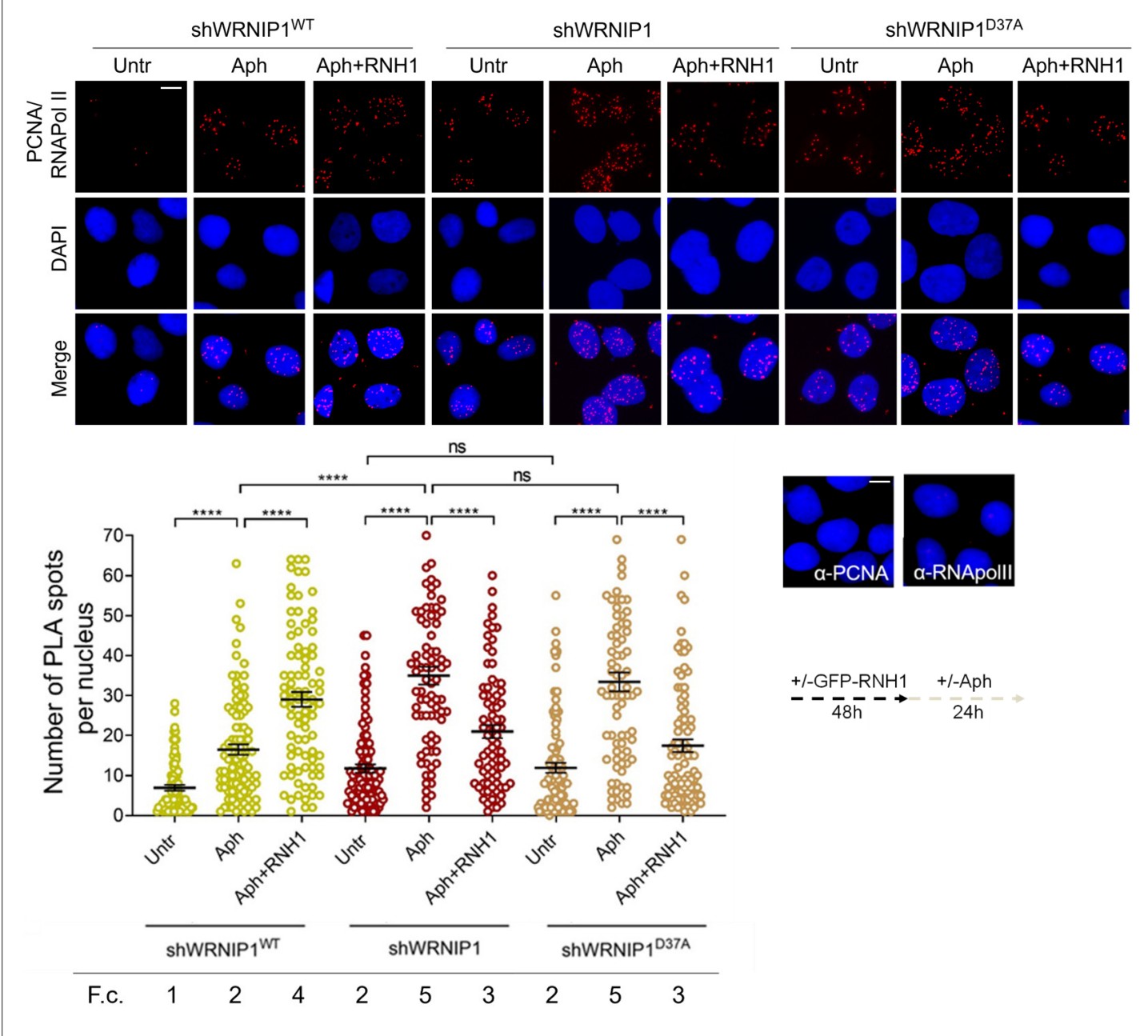

**Figure 4.** Loss of WRNIP1 or its UBZ domain promotes R-loop-dependent TRCs accumulation. Detection of TRCs by fluorescence-based PLA assay in MRC5SV, shWRNIP1 and shWRNIP1[D37A] cells. Post-transfection with GFP-tagged RNase H1 or empty vector, cells treated according to the experimental scheme. After fixing, cells were stained with antibodies against PCNA and RNA pol II. Representative images are provided for each single color channel. Each red spot represents a single interaction between proteins. No spots were observed in cells stained with each single antibody (negative control). Nuclei were counterstained with DAPI. Dot plot shows the number of PLA spots per nucleus. Data are presented as means from three independent experiments. Horizontal black lines represent the mean ± SE (ns, not significant; ****, p<0.0001; one-way ANOVA test). Fold changes (F. c) were calculated as ratio of means between each experimental sample and the relative wild-type untreated. Scale bar, 2.5 μm.

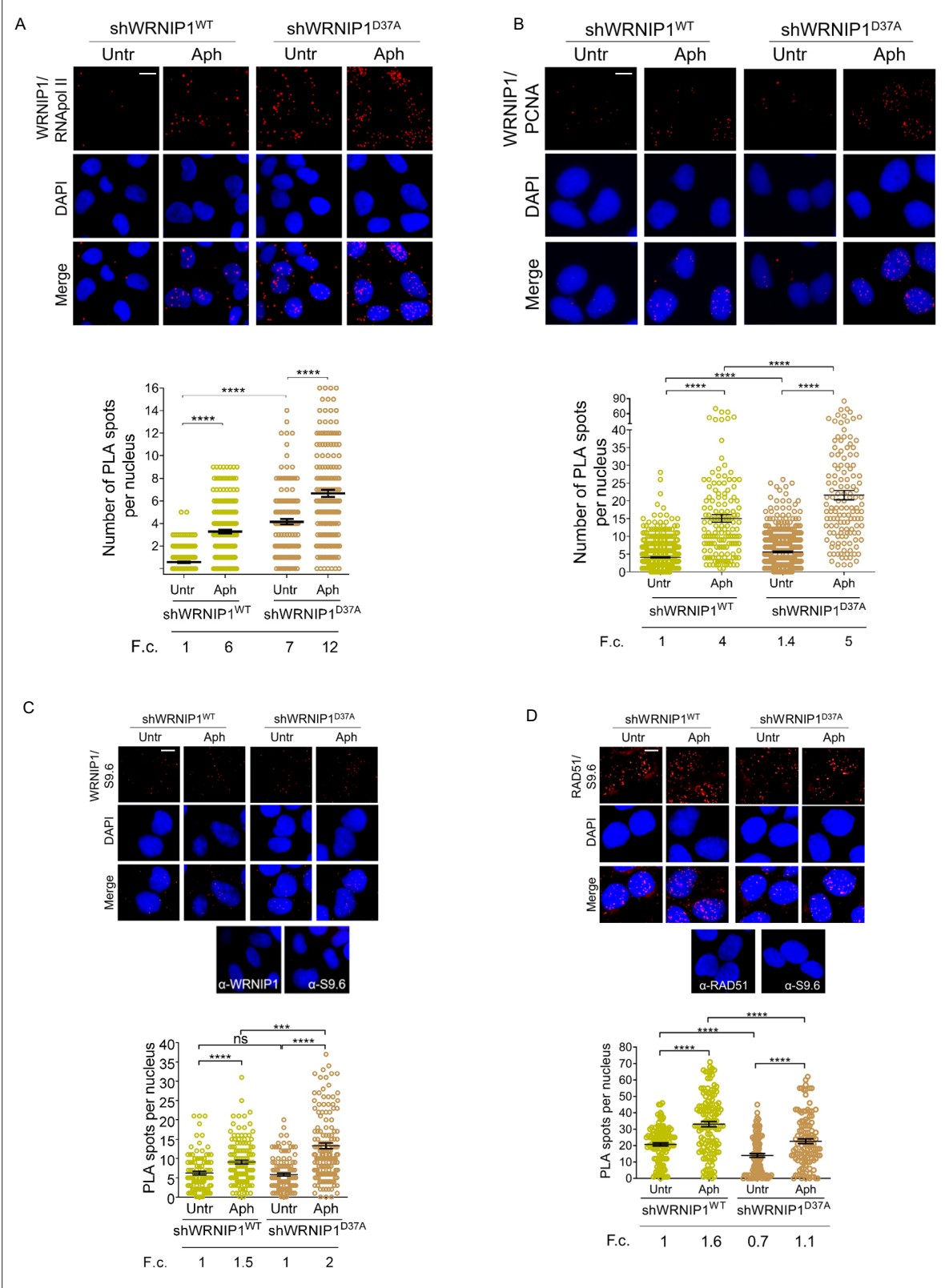

**Figure 5.** Analysis of localization of WRNIP1 or UBZ mutant by PLA assay. (**A** and **B**) Analysis of the localization of WRNIP1 near/at the transcription and replication machineries by PLA. Cells were treated or not with 0.4 μM Aph for 24 hr, fixed, and stained with antibodies against WRNIP1 and RNA pol II (**A**) or WRNIP1 and PCNA (**B**) to visualize the interaction between WRNIP1 and replication or transcription machinery, respectively. Each red spot represents a single interaction between proteins. Representative images are provided for each single color channel. Nuclei were counterstained with

*Figure 5 continued on next page*

*Figure 5 continued*

DAPI. Dot plots show the number of PLA spots per nucleus. Data are presented as means from three independent experiments. Horizontal black lines represent the mean ± SE (****, p<0.0001; one-way ANOVA test). (**C** and **D**) Detection of physical interaction between WRNIP1 and R-loops (**C**) or R-loops and RAD51(**D**). Cells were treated or not with 0.4 µM Aph for 24 hr, followed by RNase III digestion and the PLA assay. Cells were stained with antibodies against RNA-DNA hybrid (anti-S9.6) and WRNIP1 (**C**) or RNA-DNA hybrid (anti-S9.6) and RAD51 (**D**). Representative images are provided for each individual color channel. Each red spot represents a single interaction between R-loops and the respective proteins (WRNIP1 or RAD51). No spots were observed in cells stained with each single antibody (negative control). Nuclei were counterstained with DAPI. Dot plot shows the number of PLA spots per nucleus. Horizontal black lines represent the mean ± SE (ns, not significant; ***, p<0.001; ****, p<0.0001; one-way ANOVA test). Fold changes (F. c) were calculated as ratio of means between each experimental sample and the relative wild-type untreated. Scale bar, 2.5 µm.

Therefore, we concluded that in cells lacking WRNIP1 or its UBZ domain, R-loops can pose a significant impediment to replication. Additionally, the ubiquitin-binding activity of WRNIP1 may play a crucial role in restarting replication when transcription-induced fork stalling occurs.

## FA pathway is correctly activated in WRNIP1-deficient or UBZ mutant cells

Previous studies have demonstrated that a functional Fanconi anemia (FA) pathway prevents the unscheduled accumulation of transcription-associated R-loops, with FANCD2 monoubiquitination playing a critical role (*Liang et al., 2019*). Additionally, it has been reported that the FANCD2 protein complex contains WRNIP1, which, under certain conditions, facilitates the binding of the monoubiquitinated FANCD2/FANCI complex to DNA through its UBZ domain (*Socha et al., 2020*). Hence, to gain insight into defective R-loop resolution, we assessed the functional status of the FA pathway in our cells by examining FANCD2 monoubiquitination, a well-established readout of FA pathway activation (*Taniguchi et al., 2002*). To this end, we treated MRC5SV, shWRNIP1 and shWRNIP1[D37A] cells with Aph and/or DRB and performed Western blot analysis (*Figure 7A*). Our results showed that loss of WRNIP1 or its UBZ domain did not affect FANCD2 ubiquitination after treatments, indicating that the FA pathway was properly activated under our conditions (*Figure 6A*). Consistently, we observed comparable FANCD2 relocalization during R-loop-associated replication stress in all cell lines tested, and that it was reduced by transcription inhibition, providing further evidence of proper FA pathway activation (*Figure 7—figure supplement 1*). In agreement with previous findings (*Wells et al., 2022*), we found that RAD18 plays a role in FANCD2 recruitment upon MRS (*Figure 7—figure supplement 2*). Moreover, PLA assay demonstrated that, under our conditions, FANCD2 was localized near/at R-loops (anti S9.6) after Aph with a greater number of spots in shWRNIP1 and shWRNIP1[D37A] cells compared to control cells (*Figure 7B*).

We then conducted an alkaline Comet assay in MRC5SV, shWRNIP1 and shWRNIP1[D37A] cells depleted of FANCD2 to investigate the potential association between WRNIP1 and the FANCD2 pathway under replication stress. Our results revealed increased spontaneous DNA damage in both the shWRNIP1 and shWRNIP1[D37A] cells in which FANCD2 has been depleted (*Figure 7C*). Interestingly, disruption of FANCD2 led to a higher degree of DNA damage upon replication stress induced by Aph in shWRNIP1 and shWRNIP1[D37A] cells compared to control cells (*Figure 7C*).

Therefore, to explore more in-depth the role of FANCD2 in the regulation of R-loops, we examined the impact of FANCD2 depletion on the accumulation of R-loops in shWRNIP1 and shWRNIP1[D37A] cells. In agreement with our observations, the analysis of R-loop formation in WRNIP1-deficient cells depleted of FANCD2 revealed a significantly higher accumulation of R-loops in cells with a concomitant loss of both WRNIP1 and FANCD2 compared to those with a single deficiency upon MRS (*Figure 8A*). Similar results were observed in the WRNIP1 UBZ mutant cells in which FANCD2 was abrogated (*Figure 8A*).

We also performed a DNA fiber assay to evaluate restarting replication forks in shWRNIP1[WT], shWRNIP1, and shWRNIP1[D37A] cells in which FANCD2 was abrogated. Our results show that FANCD2 depletion slightly decreased the ability of the cells to restart forks from MRS (*Figure 8B*).

These findings collectively suggest that FANCD2 pathway and WRNIP1 may not be inherently interconnected.

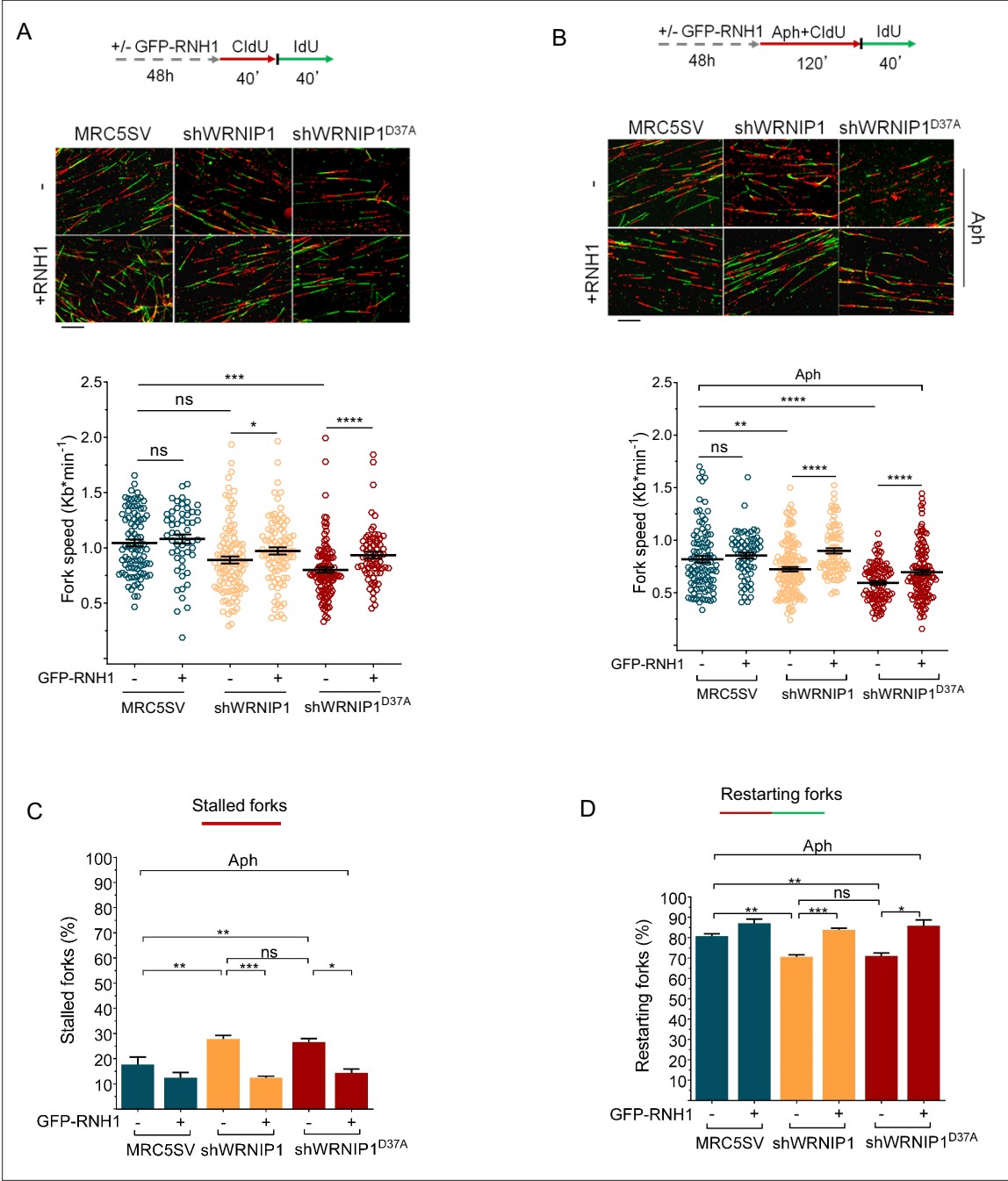

**Figure 6.** R-loops affects DNA replication in cells lacking WRNIP1 or its UBZ domain upon MRS. Experimental scheme for dual labeling of DNA fibers in MRC5SV, shWRNIP1 and shWRNIP1[D37A] cells under unperturbed conditions (**A**) or upon MRS (**B**). After transfection with GFP-tagged RNaseH1 or empty vector (-), cells were pulse-labeled with CldU, treated or not with 0.4 μM Aph, then subjected to a pulse-labeling with IdU. Representative DNA fiber images are provided. Graphs display the analysis of replication fork velocity (fork speed) in the cells. The length of the green tracks was measured. Mean values are represented as horizontal black lines (ns, not significant; *, $p < 0.05$; **, $p < 0.01$; ***, $p < 0.001$; ****, $p < 0.0001$; Mann-Whitney test). Scale bar, 10 μm. (**C** and **D**) Graphs represent the percentage of red (CldU) tracts (stalled forks) or green (IdU) tracts (restarting forks) in the cells. Error bars represent standard error (ns, not significant; *, $p < 0.05$; ** $p < 0.01$; *** $p < 0.001$; two-tailed Student t-test).

The online version of this article includes the following figure supplement(s) for figure 6:

**Figure supplement 1.** Transcription affects DNA replication in cells lacking WRNIP1 or its UBZ domain upon MRS.

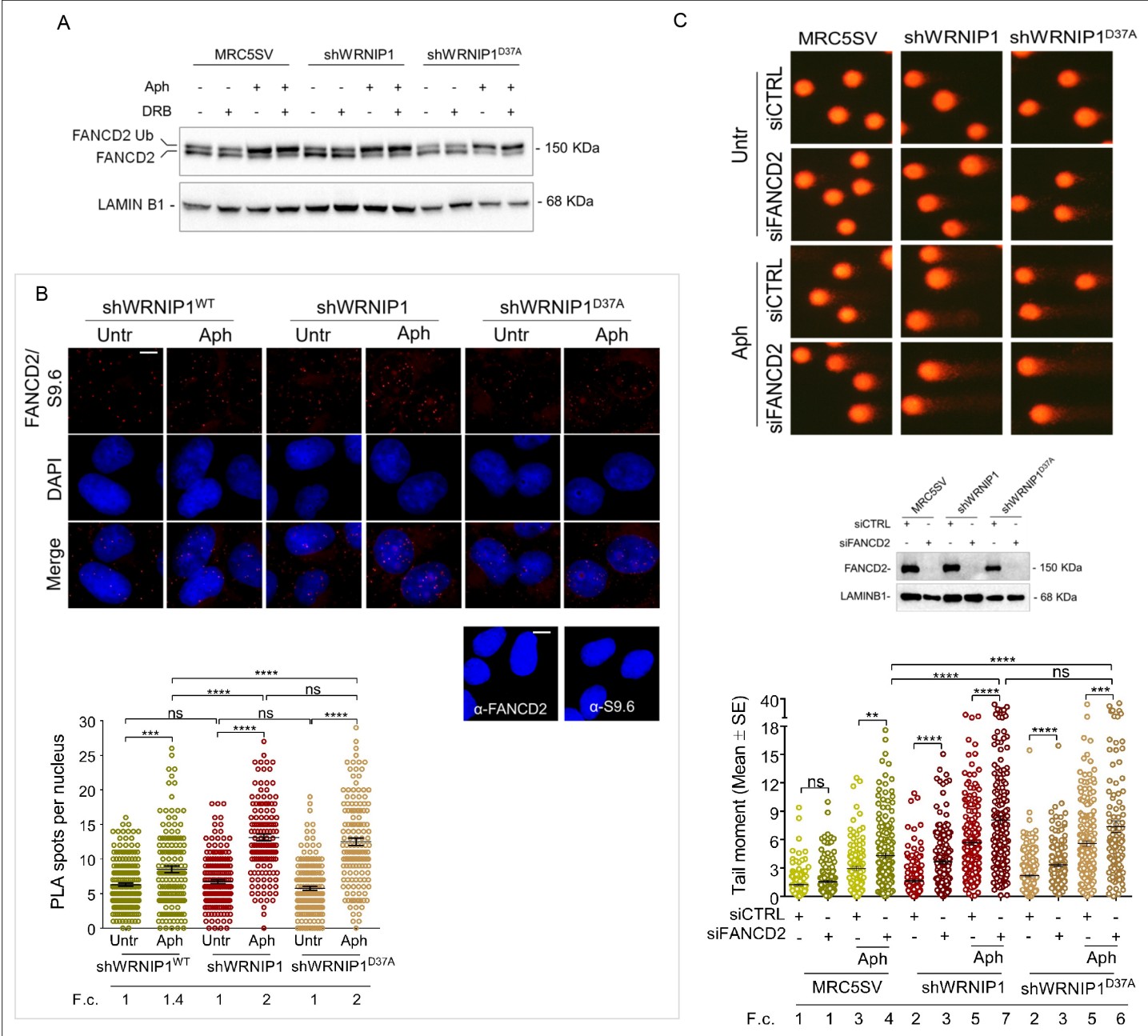

**Figure 7.** FANCD2 pathway activation in cells lacking WRNIP1 or its UBZ domain upon MRS. (**A**) Western blot analysis showing FANCD2 ubiquitination in MRC5SV, shWRNIP1 and shWRNIP1[D37A] cells. The membrane was probed with an anti-FANCD2 antibody. LAMIN B1 was used as a loading control. (**B**) Detection of the physical interaction between FANCD2 and R-loops by PLA. shWRNIP1[WT], shWRNIP1, and shWRNIP1[D37A] cells were treated or not with 0.4 μM Aph for 24 hr and subjected to RNase III digestion. Cells were stained with antibodies against RNA-DNA hybrid (anti-S9.6) and FANCD2. Representative images are provided. Each red spot represents a single interaction between R-loops and FANCD2. No spots were revealed in cells stained with each single antibody (negative control). Nuclei were counterstained with DAPI. Dot plot shows the number of PLA spots per nucleus. Data are presented as means from three independent experiments. Horizontal black lines represent the mean ±SE (ns, not significant; ***, p<0.001;****, p<0.0001; one-way ANOVA test). Scale bar, 2.5 μm. (**C**) Evaluation of DNA damage accumulation by alkaline Comet assay in MRC5SV, shWRNIP1 and shWRNIP1[D37A] transfected with control siRNAs (siCTRL) or siRNAs targeting FANCD2 (siFANCD2). After 48 hr, cells were treated or not with 0.4 μM Aph for 24 hr, then subjected to Comet assay. Dot plot shows data presented as means from three independent experiments. Horizontal black lines represent the mean (ns, not significant; ** p<0.01; *** p<0.001; **** p<0.0001; Mann-Whitney test). Representative images are provided. Western blot shows FANCD2 depletion in the cells. The membrane was probed with an anti-FANCD2, and LAMIN B1 was used as a loading control. Fold changes (F. c) were calculated as ratio of means between each experimental sample and the relative wild-type untreated.

The online version of this article includes the following source data and figure supplement(s) for figure 7:

*Figure 7 continued on next page*

*Figure 7 continued*

**Source data 1.** PDF containing original scans of the western blot (anti-FANCD2, anti-LAMINB1) and related original TIFF images.

**Source data 2.** PDF containing original scans of the western blot (anti-FANCD2, anti-LAMINB1) and related original TIFF images.

**Figure supplement 1.** Loss of WRNIP1 or its UBZ domain results in FANCD2 pathway activation upon MRS.

**Figure supplement 2.** Analysis of the dependency of FANCD2 activation on RAD18.

**Figure supplement 2—source data 1.** PDF containing original scans of the western blot (anti-RAD18, anti-LAMINB1) and related original TIFF images.

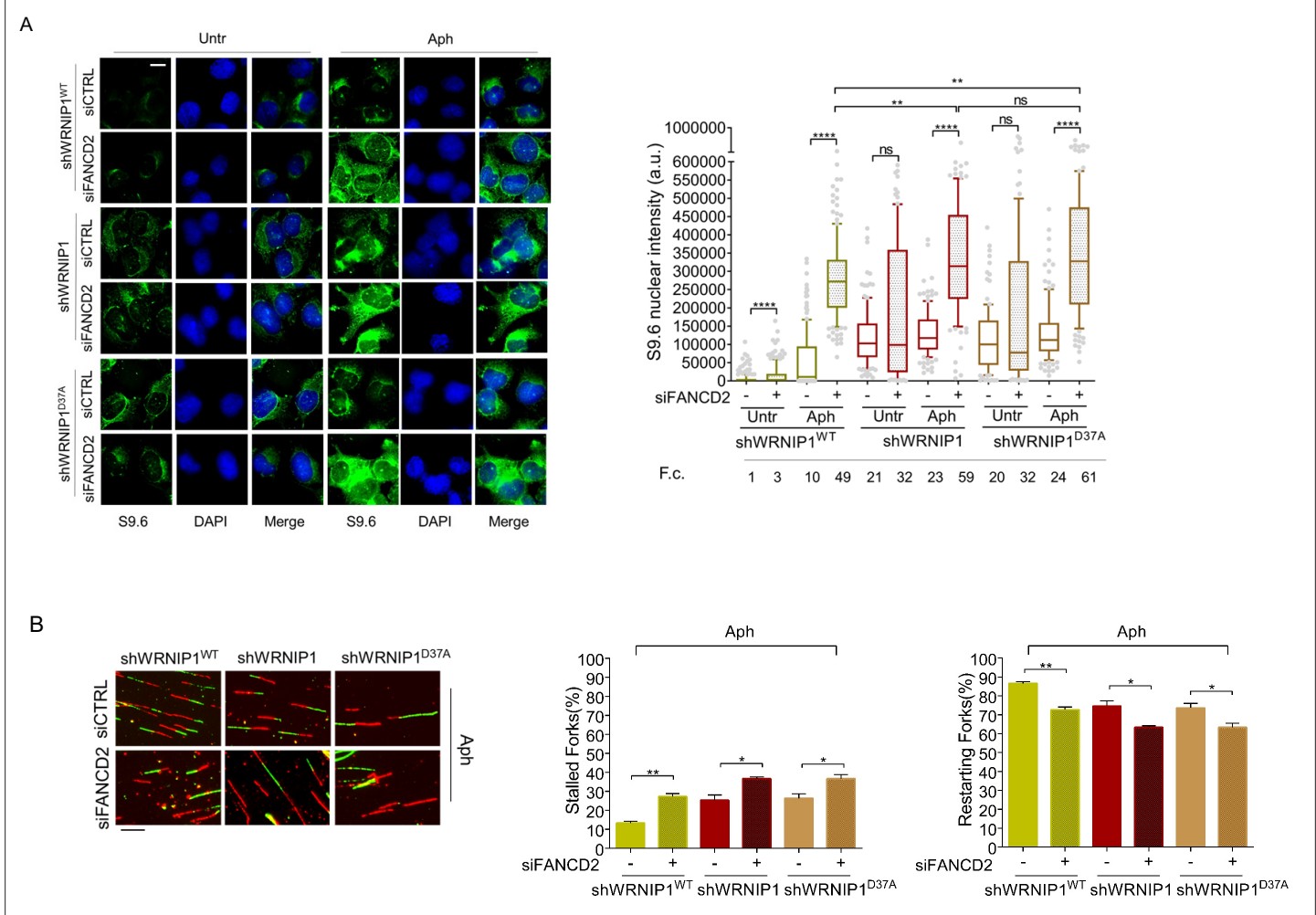

**Figure 8.** Evaluation of FANCD2 depletion on R-loop accumulation and replication dynamic. (**A**) Immunofluorescence analysis to determine R-loop levels in shWRNIP1[WT], shWRNIP1, and shWRNIP1[D37A] cells depleted or not of FANCD2 under untreated conditions or after MRS. Cells were fixed, subjected or not to RNase III digestion, and stained with anti-RNA-DNA hybrid S9.6 antibody. Representative images are provided. Nuclei were counterstained with DAPI. Dot plot shows nuclear S9.6 fluorescence intensity. Boxes and whiskers represent 20–75 and 10–90 percentiles, respectively. The line represents the median value. Data are presented as means from three independent experiments. Horizontal black lines represent the mean. Error bars represent SE (ns, not significant; **, $p<0.01$; ****, $p<0.0001$; two-tailed Student's t test). Scale bar, 2.5 μm. (**B**) Experimental scheme of dual labeling of DNA fibers in shWRNIP1[WT], shWRNIP1 and shWRNIP1[D37A] cells upon MRS. After transfection with control siRNAs (siCTRL) or targeting FANCD2 (siFANCD2), cells were pulse-labeled with CldU, treated or not with 0.4 μM Aph, then subjected to a pulse-labeling with IdU. Representative DNA fiber images are provided. Graphs represent the percentage of red (CldU) tracts (stalled forks) or green (IdU) tracts (restarting forks) in the cells. Error bars represent standard error (* $p<0.05$; ** $p<0.01$; two-tailed Student's t test). Fold changes (F. c) were calculated as ratio of means between each experimental sample and the relative wild-type untreated. Scale bar, 10 μm.

## Discussion

Discovered as one of the interactors of the WRN helicase (*Branzei et al., 2001*; *Kawabe et al., 2006*), the precise function of WRNIP1 in human cells is largely unknown. Previous results from our lab and others have revealed that WRNIP1 is crucial for the protection of HU-stalled replication forks (*Leuzzi et al., 2016*; *Porebski et al., 2019*). More recently, it has been demonstrated that checkpoint defects trigger a WRNIP1-mediated response to limit R-loop-associated genomic instability (*Marabitti et al., 2020*). Nevertheless, whether WRNIP1 is directly involved in the mechanisms of R-loop removal/resolution remains unclear.

In this study, we establish a function for WRNIP1 and its ubiquitin-binding zinc finger (UBZ) domain in counteracting DNA damage and genome instability resulting from transcription-replication conflicts (TRCs).

We have demonstrated that WRNIP1 plays a crucial role in facilitating fork restart, and this function specifically relies on its ATPase activity (*Leuzzi et al., 2016*). Notably, while the ATPase activity of WRNIP1 is essential for the restart of stalled forks (*Leuzzi et al., 2016*), it is dispensable for dealing with TRCs. In sharp contrast, a mutation disrupting the WRNIP1 UBZ domain (*Bish and Myers, 2007*; *Nomura et al., 2012*) is sufficient to induce TRCs and DNA damage at levels comparable to, if not higher than, those observed in WRNIP1-deficient cells.

The persistence of R-loops is detrimental as they can interfere with DNA replication, and R-loop-mediated fork stalling is a major feature of TRCs (*Gan et al., 2011*; *Boubakri et al., 2010*; *Wellinger et al., 2006*; *Tuduri et al., 2009*). In cells lacking WRNIP1 or expressing the WRNIP1 UBZ mutant, the elevated DNA damage and reduced fork speed are restored by RNase H1-GFP overexpression, which degrades RNA/DNA hybrids and eliminates R-loops (*Cerritelli et al., 2003*). Consistent with this, many more R-loops are detected in WRNIP1-deficient or UBZ mutant cells. This strongly suggests that unscheduled R-loops likely contribute to the increased levels of TRCs, the marked replication defects, and genomic instability observed in the absence of WRNIP1.

R-loops are the main DNA secondary structures formed during transcription and, although they regulate several physiological processes, they can also act as potent replication fork barriers (*Crossley et al., 2019*; *Gómez-González and Aguilera, 2019*). Consequently, the unscheduled accumulation of R-loops results in TRCs, leading to transcription-associated DNA damage in both yeast and human cells (*Huertas and Aguilera, 2003*; *Li and Manley, 2005*; *Santos-Pereira and Aguilera, 2015*; *Sollier et al., 2014*; *Paulsen et al., 2009*). To prevent TRCs, several replication fork protection factors are essential (*Chang and Stirling, 2017*). Disrupting of these proteins can compromise the ability of cells to overcome or resolve R-loops, leading to DNA damage. Our previous research has demonstrated that WRNIP1, not relying on its ATPase activity, protects stalled forks from nucleolytic degradation upon replication stress (*Leuzzi et al., 2016*). Additionally, WRNIP1 has been found to immunoprecipitate with BRCA2 and RAD51, and its loss destabilizes RAD51 on ssDNA (*Leuzzi et al., 2016*) Notably, R-loop formation exposes a potentially vulnerable ssDNA region on the non-template strand (*Aguilera and García-Muse, 2012*). Since BRCA2 is known to mitigate R-loop-induced DNA damage (*Shivji et al., 2018*; *Bhatia et al., 2014*; *Wang et al., 2022*), it is conceivable that WRNIP1, through its association with the BRCA2/RAD51 complex, could facilitate RAD51 recruitment or stabilization to protect the ssDNA of R-loops. Supporting this, RAD51 co-localizes with R-loops after MRS, and this co-localization depends on the WRNIP1 UBZ domain. Therefore, WRNIP1 may protect R-loops to ensure their safe handling and promote the successful restart of stalled forks. Consistent with this hypothesis, we observe WRNIP1 co-localizing with transcription/replication complexes and R-loops after MRS. It has been suggested that the incision of R-loops, a process promoting fork progression and restart at TRCs (*Chappidi et al., 2020*), requires high accuracy to prevent fork collapse and chromosomal damage, posing a risk to genome integrity (*Sollier et al., 2014*; *Promonet et al., 2020*). Our findings indicate that the R-loop-dependent accumulation of DNA damage and genome instability may suggest a preference for engaging error-prone mechanisms when WRNIP1 is depleted. Alternatively, WRNIP1 may play a role in stabilizing RAD51 on the ssDNA of DNA/RNA hybrids, facilitating fork restart through fork reversal. Indeed, one proposed mechanism for processing DNA/RNA hybrids involves the exposure of ssDNA and the loading of RAD51, which mediates fork reversal (*Stoy et al., 2023*). While these hybrids are typically resolved under physiological conditions, allowing stalled fork restart (*Stoy et al., 2023*; *Chappidi et al., 2020*), post-replicative DNA/RNA hybrids could represent pathological TRC intermediates capable of impairing replication fork progression and stimulating fork reversal.

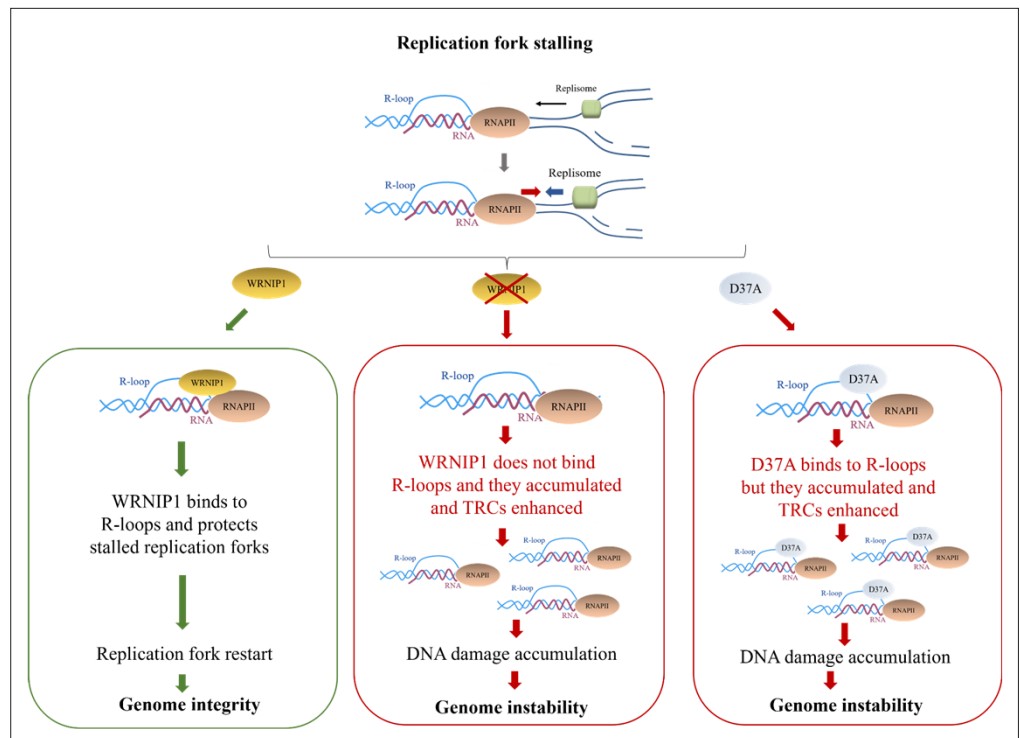

**Figure 9.** Working model for the potential role of WRNIP1 in R-loop accumulation. Upon replication fork stalling, WRNIP1 binds to R-loops and protects stalled forks, promoting genomic integrity. In the absence of WRNIP1, R-loops accumulate, and TRCs are enhanced, leading to genome instability. In case of WRNIP1 UBZ mutation, the protein binds to R-loops but they accumulate, increasing TRCs and, thus, contributing to genomic instability.

The phenotype observed in WRNIP1 UBZ mutant cells is similar to that of WRNIP1-deficient cells, suggesting that the ubiquitin-binding function of WRNIP1 could play a role in mitigating R-loop-induced TRCs. Despite the UBZ domain of WRNIP1 having an undefined function after replication perturbation (*Yoshimura et al., 2017*), human WRNIP1 in known to bind both forked DNA, resembling stalled replication forks, and template/primer DNA in an ATP-dependent fashion (*Yoshimura et al., 2009*). However, our findings indicate that WRNIP1 does not use its ATPase activity to counteract TRC-induced R-loop accumulation. Nevertheless, WRNIP1's ability to bind ubiquitin suggests its potential involvement in the metabolic processes of ubiquitinated proteins (*Bish and Myers, 2007*). The UBZ domain of WRNIP1 is implicated in the physical association with RAD18 (*Kanu et al., 2016*). RAD18-deficient cells exhibit high levels of TRCs and accumulate DNA/RNA hybrids, leading to DNA double strand breaks and replication stress (*Wells et al., 2022*). These effects are partially dependent on the failure to recruit the Fanconi anemia protein FANCD2 to R-loop-prone genomic sites (*Wells et al., 2022*). Notably, in our context, FANCD2 localizes with R-loops after MRS, and this localization is more pronounced in cells lacking WRNIP1 and its UBZ domain. Additionally, in our cell lines, although R-loop levels are more elevated in FANCD2-depleted cells, DNA damage is more pronounced in WRNIP1-deficient cells, suggesting that the FA pathway may act as a backup system to counteract TRCs. Therefore, it is tempting to speculate that the UBZ domain might contribute to directing WRNIP1 to DNA at TRC sites through RAD18, and that the elevated levels of TRCs observed in RAD18-deficient cells are due to the loss of both WRNIP1 and FANCD2 functions. This intriguing point deserves further and detailed investigation.

Altogether our findings unveil a previously unappreciated role for WRNIP1 in counteracting the pathological accumulation of transcription-associated R-loops, thereby preventing heightened genomic instability following MRS in human cells (*Figure 9*). A growing body of evidence suggests that R-loops may be implicated in the genomic instability observed in human cancer cells (*Crossley et al., 2019*; *García-Muse and Aguilera, 2019*). Notably, considering the observed overexpression of WRNIP1 in the most common human cancer types, such as lung and breast cancers, our discoveries

contribute to a deeper understanding of mechanisms cells employ to counteract TRCs. These results open avenues for exploring novel pathways that may contribute to genomic instability in cancer.

# Materials and methods

## Cell lines and culture conditions

The SV40-transformed MRC5 fibroblast cell line (MRC5SV) was a generous gift from Patricia Kannouche (IGR, Villejuif, France). MRC5SV cells stably expressing WRNIP1-targeting shRNA (shWRNIP1), and isogenic cell lines stably expressing the RNAi-resistant full-length wild-type WRNIP1 (shWRNIP1$^{WT}$) and its ATPase-dead mutant form (shWRNIP1$^{T294A}$) were generated as previously reported (*Leuzzi et al., 2016*).

Using the Neon Transfection System Kit (Invitrogen) following the manufacturer's instructions, shWRNIP1 cells were stably transfected with a plasmid expressing a FLAG-tagged full-length WRNIP1 plasmid carrying Ala substitution at Asp37 site, a missense-mutant form of WRNIP1 with dead form of ubiquitin-binding zinc finger (UBZ) domain (WRNIP1$^{D37A}$). Cells were cultured in the presence of neomycin and puromycin (1 mg/ml and 100 ng/ml, respectively) to maintain selective pressure for expression.

All cell lines were cultured in DMEM media supplemented with 10% FBS, 100 U/mL penicillin and 100 mg/mL streptomycin, and incubated at 37 °C in an humified 5% $CO_2$ atmosphere.

Prior to usage, all cell lines underwent mycoplasma contamination testing, with negative results obtained.

## Site-directed mutagenesis and cloning

Site-directed mutagenesis of the WRNIP1 full-lenght cDNA (Open Biosystems) was performed on the pCMV-FLAGWRNIP1 plasmid, which contains the wild-type ORF sequence of WRNIP1. The substitution of Asp37 to Ala in pCMV-FLAGWRNIP1 was introduced using the Quick-change XL kit (Stratagene) with mutagenic primer pairs designed according to the manufacturer's instructions. Each mutated plasmid was verified by full sequencing of the WRNIP1 ORF.

## Chemicals

Chemicals used were commercially obtained for the replication stress-inducing drug: aphidicolin (Aph; Sigma-Aldrich), hydroxyurea (HU; Sigma-Aldrich), and the transcription elongation inhibitor 5,6-dichloro-1-ß-D-ribofurosylbenzimidazole (DRB; Sigma-Aldrich). The final concentrations of the drugs used were 0.4 aphidicolin, 4 mM HU and 50 µM DRB. The transcription elongation inhibitor Cordycepin (Cordy; Sigma-Aldrich) was employed at a final concentration of 50 µM. Stock solutions for all the chemicals were prepared in DMSO at a concentration of >1000. The final concentration of DMSO in the culture medium was always < 0.1%.

## Plasmids and RNA interference

The construct used for RNase H1 overexpression experiments was generously provided by Prof. R.J. Crouch (National Institutes of Health, Bethesda, USA). As previously described (*Cerritelli et al., 2003*), the GFP-tagged RNase H1 plasmid was generated by introducing a mutation on Met27 to abrogate mitochondrial localization signal (RNase H1-M27). To express the plasmids, cells were transfected using the Neon Transfection System Kit (Invitrogen), according to the manufacturer's instructions.

For the FANCD2 genetic knockdown experiment, Interferin (Polyplus) was used according to the manufacturer's instructions with Silencer Select siRNA (Thermo Fischer Scientific) targeting the following region of the mRNA (5'-CAGCCUACCUGAGAUCCUAUT-3'). The siRNA was used at a concentration of 2.5 nM. As a control, a siRNA duplex directed against GFP was used.

The RAD18 genetic knockdown experiment was performed using Interferin (Polyplus) following the manufacturer's instructions. A Flexitube siRNA (QIAGEN) was utilized, targeting the following sequence of the mRNA (5'-ATGGTTGTTGCCCGAGGTTAA-3'), with the siRNA used at a concentration of 10 nM. As a control, a siRNA duplex directed against GFP was employed.

Depletions were confirmed by western blot using the relevant antibodies (*see below*).

## Western blot analysis

The proteins were separated on polyacrylamide gels and transferred onto a nitrocellulose membrane using the Trans-Blot Turbo Transfer System (Bio-Rad). The membranes were blocked with 5% NFDM

in TBST (50 mM Tris/HCl pH 8, 150 mM NaCl, 0.1% Tween-20) and incubated with primary antibodies for 1 hr at RT. The primary antibodies used for WB were rabbit-polyclonal anti-WRNIP1 (Bethyl Laboratories; 1:2500), mouse-monoclonal anti-FLAG (Sigma-Aldrich; 1:1000), mouse-polyclonal anti-GAPDH (Millipore; 1:5000), mouse-monoclonal anti-FANCD2 (Santa Cruz Biotecnology;1:500), rabbit-polyclonal anti-RAD18 (Abcam; 1:2000) and rabbit-polyclonal anti-LAMIN B1 (Abcam; 1:30,000).

The membranes were then incubated with horseradish peroxidase-conjugated secondary antibodies specific to the respective species (Santa Cruz Biotechnology; 1:20,000) for 1 hr at RT. Visualisation of the signal was achieved using Western Bright ECL HRP substrate (Advansta) and imaged with Chemidoc XSR+ (Chemidoc Imaging Systems, Bio-Rad).

### Chromosomal aberrations

Cells for metaphase preparations were collected following standard procedures, as previously reported (*Pirzio et al., 2008*). A cell suspension was dropped onto cold, wet slides to make chromosome preparations. After air-drying overnight, the number of breaks and gaps was observed on Giemsa-stained metaphases for each treatment condition. For each time point, a minimum of 50 chromosome metaphases were independently examined by two investigators, and chromosomal damage was scored at 100×magnification using an Olympus fluorescence microscope.

### Alkaline comet assay

DNA breakage induction was examined by alkaline Comet assay (single-cell gel electrophoresis) under denaturing conditions, as described (*Pichierri et al., 2001*). Cell DNA was stained with a fluorescent dye GelRed (Biotium) and examined at 40×magnification using an Olympus fluorescence microscope. Slides were analysed with a computerized image analysis system (CometScore, Tritek Corp). To assess the extent of DNA damage, computer-generated tail moment values (tail length ×fraction of total DNA in the tail) were used. A minimum of 200 cells were analysed for each experimental point. Apoptotic cells (identified by smaller comet head and extremely larger comet tail) were excluded from the analysis to avoid artificial enhancement of the tail moment.

### Immunofluorescence

Immunostaining for RNA-DNA hybrids was performed as described with minor changes (*Marabitti et al., 2019*). Cells were fixed in 100% methanol for 10 min at –20 °C, washed three times in PBS, pre-treated with 6 µg/ml of RNase A for 45 min at 37 °C in 10 mM Tris-HCl pH 7.5, supplemented with 0.5 M NaCl. Subsequently, cells were treated for 90 min with a 1:150 dilution of RNase III (New England Biolabs) in low-salt buffer (50 mM Tris-HCl pH 7.6, 75 mM KCl, 3 mM MgCl2, 0.1% BSA) before blocking in 2% BSA/PBS overnight at 4 °C. Cells were then incubated with the anti-DNA-RNA hybrid [S9.6] antibody (Kerafast; 1:100) overnight at 4 °C. Following each primary antibody incubation, cells were washed twice with PBS and incubated with the specific secondary antibody: goat anti-mouse Alexa Fluor-488 or goat anti-rabbit Alexa Fluor-594 (Molecular Probes). Immunostaining for dsRNA was performed as described with minor changes (*Crossley et al., 2021*). Cells were fixed in 4% PFA for 10 min at RT, washed twice with PBS, and then permeabilized in 0.25% Triton X-100 for 10 min. After two washes in PBS, samples were incubated in low-salt buffer (50 mM Tris-HCl pH 7.6, 75 mM KCl, 3 mM MgCl2, 0.1% BSA) with or without Short Cut RNase III (New England Biolabs; 1:150) for 90 min at 37 °C. Coverslips were then washed twice with PBST, followed by one wash in PBS for 5 min, before blocking in 3% BSA in PBS for 30 min. Cells were then incubated with a 1:200 dilution of anti-dsRNA monoclonal antibody J2 (Sigma-Aldrich) overnight at 4 °C. After the primary antibody, cells were washed twice with PBS and incubated with 1:200 goat anti-mouse Alexa Fluor-488 (Molecular Probes).

The incubation with secondary antibodies was carried out in a humidified chamber for 1 hr at RT. DNA was counterstained with 0.5 µg/ml DAPI.

Immunofluorescence for γ-H2AX was performed as previously described (*Murfuni et al., 2012*). Briefly, exponential growing cells were seeded onto Petri dishes, then treated (or mock-treated) as indicated. Subsequently, cells were fixed in 4% formaldehyde for 10 min, permeabilized using 0.4% Triton X-100 for 10 min and incubated with 10% FBS for 1 hr at RT. After blocking, for γ-H2AX detection, cells were incubated with the mouse-monoclonal anti-γ-H2AX primary antibody (Millipore; 1:1000) for 1 hr at RT.

FANCD2 immunostaining was performed similarly to γ-H2AX, with the exception that cells were pre-extracted with ice-cold CSK buffer for 10 min on ice. After blocking, cells were incubated with the primary anti-FANCD2 antibody (Santa Cruz Biotechnology; 1:250) for 1 hr at RT.

Following each primary antibody incubation, cells were washed twice with PBS and then incubated with the specific secondary antibody: goat anti-mouse Alexa Fluor-488 or goat anti-rabbit Alexa Fluor-594 (Molecular Probes). Secondary antibody incubation was carried out in a humidified chamber for 1 hr at RT. DNA was counterstained with 0.5 μg/ml DAPI.

Images were randomly acquired using an Eclipse 80i Nikon Fluorescence Microscope equipped with a Video Confocal (ViCo) system. For each time point, a minimum of 200 nuclei were examined. Nuclear foci were scored at a 40×magnification, and only nuclei showing more than five bright foci were considered positive. Intensity per nucleus was calculated using ImageJ. Parallel samples incubated with either the appropriate normal serum or only the secondary antibody confirmed that the observed fluorescence pattern was not attributable to artefacts.

## Dot blot analysis

Dot blot analysis was performed with minor changes according to the protocol reported elsewhere (*Morales et al., 2016*). Genomic DNA was isolated by standard extraction with Phenol/Clorophorm/Isoamylic Alcohol (pH 8.0) followed by precipitation with 3 M NaOAc and 70% Ethanol. The isolated genomic DNA was randomly fragmented for 4 hr at 37 °C with a cocktail restriction enzymes (BsrgI, EcoRI, HindIII, XbaI, Ssp1) supplemented with 1 M spermidine. After incubation, the digested DNA was cleaned up with Phenol/Clorophorm extraction and standard Ethanol precipitation. After sample quantification, 350 ng of digested DNA was incubated with RNase H overnight at 37 °C as a negative control. The same quantity of each sample was spotted onto a nitrocellulose membrane, blocked in 5% non-fat dry milk, and incubated with the anti-DNA-RNA hybrid [S9.6] antibody (Kerafast; 1:1000) or anti-dsDNA antibody (Abcam; 1:2000) overnight at 4 °C. A horseradish peroxidase-conjugated goat specie-specific secondary antibody (Santa Cruz Biotechnology) was used. Quantification on the scanned image of the blot was performed using Image Lab software with dsDNA as loading control.

## In situ PLA assay

The in situ proximity-ligation assay (PLA; Sigma-Aldrich) was performed according to the manufacturer's instructions. Exponential growing cells were seeded into 8-well chamber slides (Nun Lab-Tek) at a density of 1.5–2.5×10$^4$ cells/well. After the indicated treatment, cells were permeabilized with 0.5% Triton X-100 for 10 min at 4 °C, fixed with 3% formaldehyde/ 2% sucrose solution for 10 min, and then blocked in 3% BSA/PBS for 15 min. After washing with PBS, cells were incubated with the two relevant primary antibodies.

For the detection of R-loops using PLA with the anti-S9.6 antibody, cells were fixed with ice-cold methanol for 10 min and treated with 6 μg/ml of RNase A for 45 min at 37 °C in 10 mM Tris-HCl pH 7.5, supplemented with 0.5 M NaCl. Subsequently, cells were treated with a 1:150 dilution of RNase III (New England Biolabs) in low-salt buffer (50 mM Tris-HCl pH 7.6, 75 mM KCl, 3 mM MgCl2, 0.1% BSA) for 90 min before blocking in 2% BSA/PBS for 1 h at 37 °C.

The primary antibodies used were as follows: anti-FLAG (mouse-monoclonal, Sigma-Aldrich; 1:250), anti-WRNIP1 (rabbit-polyclonal, Bethyl; 1:500), anti-S9.6 (mouse-monoclonal, Kerafast; 1:200), anti-PCNA (rabbit-polyclonal, Abcam; 1:500), anti-RNA pol II (mouse-monoclonal, Santa Cruz Biotechnology; 1:200), recombinant anti-S9.6 (rabbit-monoclonal, Kerafast; 1:200), anti-RAD51 (rabbit-polyclonal, Abcam; 1:300) and anti-FANCD2 (mouse-monoclonal, Santa Cruz Biotechnology; 1:100). The negative control consisted of using only one primary antibody.

Samples were incubated with secondary antibodies conjugated with PLA probes MINUS and PLUS: the PLA Probe anti-Mouse PLUS and anti-Rabbit Minus (Sigma-Aldrich). The incubation with all antibodies was carried out in a humidified chamber for 1 hr at 37 °C. Next, the PLA probes MINUS and PLUS were ligated using their connecting oligonucleotides to produce a template for rolling-cycle amplification. During amplification, the products were hybridized with a red fluorescence-labeled oligonucleotide. Samples were mounted in Prolong Gold antifade reagent with DAPI (blue). Images were acquired randomly using an Eclipse 80i Nikon Fluorescence Microscope equipped with a Video Confocal (ViCo) system.

## DNA fiber analysis

Cells were pulse-labeled with 50 µM 5-chloro-2'-deoxyuridine (CldU) and 250 µM 5-iodo-2'- deoxyuridine (IdU) at specified times, with or without treatment as outlined in the experimental schemes. DNA fibres were prepared and spread out as previously described (*Leuzzi et al., 2016*). For the immunodetection of labeled tracks, the following primary antibodies were used: anti-CldU (rat-monoclonal anti-BrdU/CldU, BU1/75 ICR1 Abcam; 1:100) and anti-IdU (mouse-monoclonal anti-BrdU/IdU, clone b44 Becton Dickinson; 1:10). The secondary antibodies were goat anti-mouse Alexa Fluor 488 or goat anti-rabbit Alexa Fluor 594 (Molecular Probes; 1:200). The incubation with antibodies was carried out in a humidified chamber for 1 hr at RT.

Images were randomly acquired from fields with untangled fibres using Eclipse 80i Nikon Fluorescence Microscope, equipped with a Video Confocal (ViCo) system. The length of green-labeled tracks was measured using the Image-J software, and values were converted into kilobases using the conversion factor 1 µm=2.59 kb as reported (*Basile et al., 2014*). A minimum of 100 individual fibres were analysed for each experiment, and the mean of at least three independent experiments is presented. Statistics were calculated using Graph Pad Prism Software.

## Statistical analysis

Statistical analysis was performed using Prism 8 (GraphPad Software). Details of the individual statistical tests are indicated in the figure legends and results. Statistical differences in all case were determined by Student's t-test, Mann-Whitney test, or one-way ANOVA test. In all cases, not significant: $p > 0.05$; * $p < 0.05$; ** $p < 0.01$; *** $p < 0.001$; **** $p < 0.0001$. All the experiments were repeated at least three times unless otherwise noted. Fold changes (F. c.) were calculated as ratio of means between each experimental sample and the relative wild-type untreated.

---

## Additional information

### Funding

| Funder | Grant reference number | Author |
| --- | --- | --- |
| Fondazione AIRC per la ricerca sul cancro ETS | Investigator grant 19971 | Annapaola Franchitto |
| Fondazione AIRC per la ricerca sul cancro ETS | Investigator grant 17383 | Pietro Pichierri |

The funders had no role in study design, data collection and interpretation, or the decision to submit the work for publication.

### Author contributions

Pasquale Valenzisi, Data curation, Formal analysis, Investigation, Writing – original draft, Writing – review and editing; Veronica Marabitti, Data curation, Investigation, Writing – review and editing; Pietro Pichierri, Conceptualization, Supervision, Funding acquisition, Writing – original draft, Writing – review and editing; Annapaola Franchitto, Conceptualization, Data curation, Supervision, Funding acquisition, Writing – original draft, Project administration, Writing – review and editing

### Author ORCIDs

Pietro Pichierri http://orcid.org/0000-0002-2702-3523
Annapaola Franchitto http://orcid.org/0000-0003-4232-4727

Reviewer #1 (Public Review): https://doi.org/10.7554/eLife.89981.3.sa1
Reviewer #2 (Public Review): https://doi.org/10.7554/eLife.89981.3.sa2
Reviewer #3 (Public Review): https://doi.org/10.7554/eLife.89981.3.sa3
Author Response https://doi.org/10.7554/eLife.89981.3.sa4

### Data availability

All data generated or analysed during this study are included in the manuscript and supporting files.

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
