## [Editor Report · eLife assessment]

This **valuable** paper examines the role of the WRNIP1 AAA+ ATPase in regulating R-loop formation, which induces a conflict with active replication forks and transcription. The authors provide **convincing** evidence to support a role of the ubiquitin-binding UBZ domain of WRNIP1 in R-loop suppression generated by this conflict. The work is of interest to researchers who work on genome stability/instability.

---

## [Referee Report · Reviewer #1 (Public Review)]

This paper describes the role of WRNIP1 AAA+ ATPase, particularly its UBZ domain for ubiquitin-binding, but not ATPase, to prevent the formation of the R-loop when DNA replication is mildly perturbated. By combining cytological analysis for DNA damage, R-loop and chromosome aberration with the proximity ligation assay for colocalization of various proteins involved in DNA replication and transcription, the authors provide solid evidence to support the claim. The authors also revealed a distinct role of WRNIP1 in the prevention of R-loop-induced DNA damage from FANCD2, which is inconsistent with the known relationship between WRNIP1 and FANCD2 in the repair of crosslinks.

---

## [Referee Report · Reviewer #2 (Public Review)]

This paper aims at establishing the role of WRN-interacting protein 1 (WRNIP1) and its UBZ domain (an N-terminal ubiquitin-binding zinc finger domain) on genome instability caused by mild inhibition of DNA synthesis by aphidicolin. The authors used human MRC5 fibroblasts investigated with standard methods in the field. The results clearly showed that WRNIP1 silencing and UBZ-mutation (D37A) increased DNA damage, chromosome aberrations, and transcription-replication conflicts caused by aphidicolin.

The conclusions of the paper are overall well supported by results, however, aspects of some data analyses would need to be clarified and/or extended.

(1) The methods (immunofluorescence microscopy and dot-blots) to determine R-loop levels can lack sensitivity and specificity. In particular, since the S9.6 antibody can bind to other structures besides heteroduplex, dot-blot analyses only grossly assess R-loop levels in cellular samples of purified nucleic acids, which are constituted by many different types of DNA/RNA structures.

(2) Experimental plan has analyzed the impact of WRNIP1 lack or mutations at steady-state conditions. Thus, the possible role of WRNIP1 at an early step of the mechanism would require some sort of kinetics analysis of the molecular process, therefore not at steady-state conditions. The findings of a co-localization of R-loops and WRNIP1 have been obtained with the S9.6 antibody, which recognizes DNA-RNA heteroduplexes. Since WRNIP1 is known to be recruited at stalled forks and DNA cleavage sites, it is not surprising that WRNIP1 is very close to heteroduplexes, abundant structures at replication forks and cleavage sites. Similar interpretations may also be valid for Rad51/S9.6 co-localization findings.

(3) Determination of DNA damage, chromosome aberration, and co-localization data are reported as means of measurements with appropriate statistics. However, the fold-change values relative to corresponding untreated samples are not reported. In some instances, it seems that WRNIP1 silencing or mutations actually reduce or do not affect aphidicolin effects. That leaves open the interpretation of specific results.

---

## [Referee Report · Reviewer #3 (Public Review)]

Summary:

In the manuscript by Valenzisi et al., the authors report on the role of WRNIP1 to prevent R-loop and TRC-associated DNA damage. The authors claim WRNIP1 localizes to TRCs in response to replication stress and prevents R-loop accumulation, TRC formation, replication fork stalling, and subsequent DNA damage. While the findings are of potential significance to the field, the strength of evidence in support of the conclusions is lacking.

In the revised submission by Velenzisi et al., the manuscript is still missing the controls that were requested in the original review. One cannot conclude the D37A mutant is unable to rescue DNA damage unless it is shown in the same experiment that the WT is able to rescue it. This is also true for the fork speed, stalled forks, and restarting forks experiments. Below is a list of Figures missing key controls.

Figure 1B -missing the shWRNIP1WT control

Figure 1C - missing the MRC5SV control

Figure 1D - missing the shWRNIP1WT control

Figure 3C - missing the shWRNIP1WT control

Figure 5A - missing the shWRNIP1WT control

Figure 5B - missing the shWRNIP1WT control

Figure 5C - missing the shWRNIP1WT control

Figure 5D - missing the shWRNIP1WT control

Figure 6C - missing the shWRNIP1WT control

Also, the authors did not explain the result showing shWRNIP1 decreases DNA damage compared to MRC5SV in Figure 1D (compare lanes 4 and 8). Again, this suggests WRNIP1 actually increases DNA damage in response to Aph and DRB. This concern was raised in the original peer review, and it remains unaddressed in the revised manuscript.

The use of RNaseHIII increases the specificity of the S9.6 antibody and improves confidence in the DNA-RNA hybrid data throughout the manuscript.

---

## [Author Response]

The following is the authors’ response to the original reviews.

**Reviewer #1:**
This paper describes the role of WRNIP1 AAA+ ATPase, particularly its UBZ domain for ubiquitinbinding, but not ATPase, to prevent the formation of the R-loop when DNA replication is mildly perturbated. By combining cytological analysis for DNA damage, R-loop, and chromosome aberration with the proximity ligation assay for colocalization of various proteins involved in DNA replication and transcription, the authors provide solid evidence to support the claim. The authors also revealed a distinct role of WRNIP1 in the prevention of R-loop-induced DNA damage from FANCD2, which is inconsistent with the known relationship between WRNIP1 and FANCD2 in the repair of crosslinks.One concern is the relationship between WRNIP1 and FANCD2 (Figure 6) in the suppression of Rloop-induced DNA damage. This is different from the relationship in inter-crosslink (ICL) repair (Socha et al. 2020), which shows the epistatic relationship between WRNIP1 as well as its UBZ domain and FANCD2 in the ICL repair. The authors need to re-evaluate the role of FNACD2 in Rloop suppression under mild replication stress (MRS) by analyzing R-loop formation in the FANCD2 knockdown (KD) cells as well as colocalization of FANCD2 with PCNA and RNA polymerase II by the PLA method and restarting the forks by the DNA coming.In this line, it is important to show PLA signal between FANCD2 and R-loop depends on WRNIP1 since WRINP1 recruits FANCD2 in ICL repair (Socha et al. 2020).

In the study referenced by the reviewer, the authors implicated WRNIP1 in repairing interstrand crosslinks (ICLs) induced by agents, such as TMP/UVA, MMC, and Cisplatin (Socha et al., 2020). For the repair of ICLs, the FANCD2/FANCI complex, the central component of the FA pathway, must be recruited to DNA. The study suggests a potential role for WRNIP1 in loading the FANCD2/FANCI complex onto DNA immediately after ICL formation. However, even in the absence of WRNIP1, a residual recruitment of the FANCD2/FANCI complex to DNA was observed, possibly due to alternative mechanisms, as proposed by the authors. Interestingly, the study did not establish a similar relationship between WRNIP1 and FANCD2 after treatments that does not induce ICLs, demonstrating that WRNIP1 and FANCD2 may also play independent roles. Hence, our data demonstrating a distinct role of WRNIP1 from the FA pathway in response to R-loop-associated replication stress are not inconsistent with prior findings. Additionally, considering the UBZ domain ability to interact with ubiquitin in both its free form and when conjugated to other proteins, thereby regulating protein functions, it is not surprising that the UBZ domain of WRNIP1 may also play a role in the response to R-loop accumulation.

Therefore, to address the reviewer's request for a more in-depth exploration of the role of FANCD2 in the regulation of R-loops, we chose to examine the impact of FANCD2 loss on the accumulation of R-loops in WRNIP1-deficient and WRNIP1 UBZ mutant cells, as well as on the dynamics of stalled forks following aphidicolin-induced MRS. Additionally, we investigated the colocalization between FANCD2 and R-loops in shWRNIP1WT, shWRNIP1 and shWRNIP1D37A cells. Details are provided below.

In agreement with our observations, the analysis of R-loop formation upon MRS, in WRNIP1deficient cells depleted of FANCD2, revealed a significantly higher accumulation of R-loops in cells with a concomitant loss of both WRNIP1 and FANCD2 compared to those with a single deficiency (see Fig. 6D of the revised manuscript). Similar results were observed in the WRNIP1 UBZ mutant cells in which FANCD2 was abrogated (see Fig. 6D of the revised manuscript). It is important to note that, to eliminate contaminant-free RNA, particularly dsRNA, which could interfere with the binding of RNA-DNA hybrids by the S9.6 antibody (Hartono et al., 2018), and to determine the proximity between FANCD2 and R-loops more accurately, cells were treated with RNase III, following established protocols (Crossley et al., 2020).

Furthermore, we examined the interaction of FANCD2 with R-loops using a proximity ligation assay (PLA). Our findings revealed significant colocalization between FANCD2 and R-loops in the absence of WRNIP1 and in WRNIP1 UBZ mutant cells following low-dose aphidicolin treatment and RNase III exposure, showing a significant increase compared to the control counterpart (shWRNIP1WT cells; see Fig. 6B of the revised manuscript). Consequently, we conclude that neither WRNIP1 nor its UBZ domain is necessary for FANCD2 recruitment under conditions of MRS.

We also performed a DNA fiber assay to evaluate restarting replication forks in shWRNIP1WT, shWRNIP1 and shWRNIP1D37A cells in which FANCD2 was abrogated. Our results show that FANCD2 depletion slightly decreased the ability of the cells to restart forks from MRS (see Fig. 6E of the revised manuscript).

Given a low number (2-4) of PLA foci for WRNIP1-RNA polymerase II or WRNIP1 and R-loop (Figure 4B and 4D), how does this colocalization reflect the functional significance?

The data from the PLA of Figures 4B and 4D are reported as the mean of three independent experiments. It is important to note that we have introduced a new Figure 4D. To selectively assess R-loop structures, cells were treated with RNase III, a double-stranded RNA-specific endoribonuclease, following established protocols (Crossley et al., 2020). Our PLA analysis confirms the localization of WRNIP1 at/near R-loops in shWRNIP1 and shWRNIP1D37A cells, and this phenomenon is more evident in WRNIP1 UBZ mutant cells (see Fig. 4D of the revised manuscript). Specifically, the new protocol allows us to visualize a higher number of PLA foci, and we observed that Aph increased the spots per nucleus in shWRNIP1D37A cells compared to the previous experiment.

Regarding the Fig. 4B, it is not uncommon for a low number of PLA spots per nucleus to correspond to a phenotypic effect. For instance, a similar low average in the colocalization of PCNA or RNA pol II with FANCD2 has been observed in a prior paper as well, suggesting that transcription-replication collisions occur upon Aph-induced MRS (Okamoto et al., 2019). Also, not all R-loops could be “targeted” by WRNIP1.

It would be helpful to readers if the authors were to provide a summary figure of this paper.

As suggested by the reviewer, we have developed a model to summarize the findings obtained in our study (see Fig. 6F of the revised manuscript).

Minor points:(1) Most of the cytological images in the paper show only colocalized ones, which makes it hard to see a signal. Please show a single-color image.

For a better visualization of nuclei signals in the figures, single-color images have been provided for Figs. 2A; 3B; 4A, B, C, D and E; 6B and D; Suppl. Fig. 2A and B of the revised manuscript.

(2) In Figure 2A, only one or two S9.6 focus(foci) can be seen. Why 1 or 2? This focus marks a specific chromosomal locus such as the centromere or telomere.

We agree with the reviewer that the observed foci in nuclei may indicate a specific chromosomal locus, such as telomeres or centromeres.

(3) Figure 3A, graph: Why this graph does not use a dot plot like Figure 1B and Figure 3C?

The graph in Figure 3A has been represented as a dot plot, as requested.

(4) Figure 1C: P values between unperturbed conditions should be provided.

In Figure 1C, P values comparing unperturbed conditions were already included. The results showed no significance between shWRNIP1 and shWRNIP1D37A cells when compared to MRC5SV cells and, similarly, to shWRNIP1T294A cells, as indicated in the corresponding legend.

(5) Figure 2B: Please provide the quantification or show the reproducibility of the data.

The quantification of R-loops using the S9.6 monoclonal antibody is not accurate, as the specificity for RNA-DNA hybrids is questionable (Hartono et al., 2018). Therefore, to demonstrate the reproducibility of the findings in Fig. 2B, we conducted a repeat of the dot-blot experiment. We treated the samples with RNase H to degrade RNA-DNA hybrids and hybridized the membrane with an anti-dsDNA to quantify R-loop levels more accurately. Our analysis confirms that the S9.6 signal strongly accumulates in shWRNIP1 cells compared to shWRNIP1WT cells (see Fig. 2B of the revised manuscript). Additionally, a graph illustrating the fold-change values of the S9.6/dsDNA signal relative to wild-type untreated cells is provided.

(6) Figure 4A: the expression of RNaseH under aphidicolin addition increased colocalization of PCNA and RNA pol II. It is important to mention the result and provide an explanation of why it is increasing in the main text.

Although the result may appear unexpected, and we lack experiments that explain the nature of this phenotype, a previous study reported that overexpression of RNase H1 in mammalian cells may lead to a dose-dependent reduction of certain proteins of the repair pathway, resulting in a significant accumulation of DNA damage (Shen et al., 2017). Consequently, the observed increase in TRCs upon RNase H1 overexpression in wild-type cells may be attributed to the disruption of proteins that, by impairing the repair process, can potentially cause more fork stalling and, consequently, more conflicts. We have introduced a comment in the text.

**Reviewer #2:**
This paper aims at establishing the role of WRN-interacting protein 1 (WRNIP1) and its UBZ domain (an N-terminal ubiquitin-binding zinc finger domain) on genome instability caused by mild inhibition of DNA synthesis by aphidicolin. The authors used human MRC5 fibroblasts investigated with standard methods in the field. The results clearly showed that WRNIP1 silencing and UBZ-mutation (D37A) increased DNA damage, chromosome aberrations, and transcription-replication conflicts caused by aphidicolin. The conclusions of the paper are overall well supported by results, however, aspects of some data analyses would need to be clarified and/or extended.(1) The methods (immunofluorescence microscopy and dot-blots) to determine R-loop levels can lack sensitivity and specificity. In particular, since the S9.6 antibody can bind to other structures besides heteroduplex, dot-blot analyses only grossly assess R-loop levels in cellular samples of purified nucleic acids, which are constituted by many different types of DNA/RNA structures.

To eliminate contaminant-free RNA, particularly dsRNA, which could interfere with the capture of RNA-DNA hybrids by the S9.6 antibody (Hartono et al., 2018), and to determine R-loop levels more accurately, we treated cells with RNase III, following established protocols (Crossley et al., 2020). Under our experimental conditions, RNase III treatment significantly reduced the amount of dsRNA, nearly eliminating it, as evaluated using a specific antibody against dsRNA (see Suppl Fig 2 of the revised manuscript). To better appreciate the effect of the loss of WRNIP1 or its UBZ domain on Rloop accumulation and the amount of DNA damage, we have reproduced key data (see Figs 2B; 3B; 4D and E; 6B of the revised manuscript). Our analysis from immunofluorescence experiments, performed using a dsRNA ribonuclease (RNase III), confirms higher R-loop accumulation in WRNIP1-deficient or WRNIP1 UBZ mutant cells compared to control cells (Fig 3B). Additionally, proximity ligation assay (PLA) data are consistent with those previously presented and, in some cases, are more readily interpretable (see Figs 4D and E; 6B of the revised manuscript). Finally, we performed a new dot-blot experiment (see Fig. 2B of the revised manuscript). We treated with RNase H to degrade RNA-DNA hybrids and hybridized the membrane with an anti-dsDNA antibody to quantify R-loop levels more accurately. Our analysis confirms a significant accumulation of the S9.6 signal in shWRNIP1 cells compared to shWRNIP1WT cells. Additionally, a graph illustrating the foldchange values of the S9.6/dsDNA signal relative to wild-type untreated cells is provided.

(2) Experimental plan has analyzed the impact of WRNIP1 lack or mutations at steady-state conditions. Thus, the possible role of WRNIP1 at an early step of the mechanism would require some sort of kinetics analysis of the molecular process, therefore not at steady-state conditions. The findings of a co-localization of R-loops and WRNIP1 have been obtained with the S9.6 antibody, which recognizes DNA-RNA heteroduplexes. Since WRNIP1 is known to be recruited at stalled forks and DNA cleavage sites, it is not surprising that WRNIP1 is very close to heteroduplexes, abundant structures at replication forks and cleavage sites. Similar interpretations may also be valid for Rad51/S9.6 co-localization findings.

Investigating the potential role of WRNIP1 at an early step in the mechanism is undoubtedly very interesting and requires separate investigation. Our decision to explore the relevance of the loss of WRNIP1 or WRNIP1 mutations under steady-state conditions is based on a preliminary alkaline comet assay (provided below). The comet assay, performed at various exposure times of aphidicolin at a concentration of 0.4 micromolar, clearly indicates that the most significant effect on DNA damage accumulation in WRNIP1-deficient cells occurs after 24 hours of treatment. Therefore, we have chosen to study the transcription-associated genomic instability in our cells by treating them with a low-dose of aphidicolin for 24 hours to maximize the effect.

**Author response image 1. sa4fig1:** 

We agree that the presence of WRNIP1 or RAD51 in proximity to R-loops is consistent with their roles and may not be surprising. However, these experiments formally demonstrate their proximity to R-loops under our conditions. Notably, the new graphs, obtained from experiments repeated by treating with RNase III to reduce the amount of dsRNA and improve the specificity of the S9.6 antibody, show increased interaction of the mutated form of WRNIP1 in the UBZ domain with Rloops when compared to the wild-type form. Additionally, it is more evident that the presence of RAD51 at/near R-loops is reduced in WRNIP1 UBZ mutant cells both in untreated conditions and after MRS (see Figs 4D and E of the revised manuscript).

(3) Determination of DNA damage, chromosome aberration, and co-localization data are reported as means of measurements with appropriate statistics. However, the fold-change values relative to corresponding untreated samples are not reported. In some instances, it seems that WRNIP1 silencing or mutations actually reduce or do not affect aphidicolin effects. That leaves open the interpretation of specific results.

To better evaluate the significance of the data presented in the study, we have introduced the foldchange values calculated with respect to the untreated samples, as requested by the reviewer. This allowed us to conclude that the loss of WRNIP1 or the expression of the UBZ mutant form of WRNIP1 does not reduce in any case the effects of aphidicolin-induced mild replication stress.

I would suggest some additional experiments or analyses to get more convincing results:(1) DNA damage should be verified also with other methods, such as DNA damage markers pH2AX and 53BP1.

The quantification of DNA damage was also corroborated by determining the percentage of gammaH2AX-positive cells, as reported in Supplementary Figure 1B. This result is consistent with the findings from the comet assay, confirming transcription-dependent DNA accumulation in shWRNIP1 and shWRNIP1D37A cells. Regarding the 53BP1 marker, we believe that the existing data sufficiently demonstrate DNA damage accumulation in the absence of WRNIP1 or when its UBZ domain is mutated, providing comprehensive support to the study without necessitating additional results.

(2) Repair foci may also be detected with Rad51 foci. That will also provide evidence for increased DNA damage levels under the tested conditions.

Our prior study identified WRNIP1 as a crucial factor for RAD51 function (Leuzzi et al., 2016). Loss of WRNIP1 indeed results in a defective relocalization of RAD51 to chromatin. Consequently, the analysis of RAD51 foci may be not a useful readout to evaluate DNA damage levels under our conditions.

(3) WRNIP1 effects should be presented as FC (fold-changes) of DNA damage, PLA results, chromosomal errors, etc, to provide evidence of the level of effects on the tested phenotypes.

We have introduced the fold-change values calculated with respect to the untreated samples, as requested by the reviewer, for a more comprehensive analysis in the graph of Figs. 1B, C and D; 2A and B; 3A, B and C; 4A, B, C, D and E; 6B, C and D.

(4) R-loop detection ideally should be performed by one of the several types of immunoprecipitation techniques. Alternatively, dot-blot assays should be performed with a 1:2 dilution series of each sample. Then, heteroduplexes should be detected with S9.6 along with a general aspecific dye for DNA quantity in each spot. Next, densitometric analyses of S9.6 signal should be normalized over DNA quantity.

We acknowledge that the quantification of R-loops using the S9.6 monoclonal antibody is not accurate, as the specificity for RNA-DNA hybrids is questionable (Hartono et al., 2018). Therefore, to overcome this issue, we repeated the experiment shown in Fig. 2B. We treated the samples with RNase H to degrade RNA-DNA hybrids and hybridized the membrane with an anti-dsDNA antibody to quantify R-loop levels more accurately. Our analysis confirms that the S9.6 signal strongly accumulates in shWRNIP1 cells compared to shWRNIP1WT cells (see Fig. 2B of the revised manuscript). Additionally, a graph illustrating the fold-change values of the S9.6/dsDNA signal relative to wild-type untreated cells is provided.

(5) A major focus on WRNIP1 D37A and T294A mutations may also make the paper overall more convincing. For instance: do the mutations affect protein recruitment at damaged chromatin? Do they increase repair foci? Do they affect the recruitment of WRN or BLM helicases or specific nucleases at chromatin under the tested conditions of MRS?

To address this point raised by the reviewer, we performed a chromatin experiment to assess the ability of WRNIP1 and its mutated forms to translocate to chromatin upon MRS. Our analysis shows that the mutated forms of WRNIP1 do not exhibit any defects in recruitment to chromatin, although the levels of the WRNIP1 ATPase mutant appear lower than the others (see Western blotting provided below for the reviewer’s use only, Fig. A). Additionally, we tested the presence of WRN helicase, which does not show any difference between cells lines (see Western blot provided below, Author response image 2).

**Author response image 2. sa4fig2:** 

(6) I suggest revising the text for spelling errors.

The manuscript has been carefully revised to identify and correct any spelling errors that may have occurred.

**Reviewer #3:**
In the manuscript by Valenzisi et al., the authors report on the role of WRNIP1 to prevent R-loop and TRC-associated DNA damage. The authors claim WRNIP1 localizes to TRCs in response to replication stress and prevents R-loop accumulation, TRC formation, replication fork stalling, and subsequent DNA damage. While the findings are of potential significance to the field, the strength of evidence in support of the conclusions is lacking.Weaknesses:(1) The authors fail to utilize the proper controls throughout the manuscript in regard to the shWRNIP1, WT, and mutant cell lines. It is unclear why the authors failed to use the shWRNIP1WT line in the comet assay, DNA fiber assay, and the FANCD2 assays. This is a key control for (i) the use of only a single shRNA (most studies will use at least 2 different shRNAs) and (ii) the use of the mutant WRNIP1 lines. In several figures, the authors only show the effect of the UBZ mutant, but don't include the ATPase mutant or WT for comparison. Including these is essential.

We agree with the reviewer's criticism that the use of shWRNIP1WT cells as a control is more appropriate. Therefore, all the new experiments presented in the revised version of the manuscript have been performed using the shWRNIP1WT cells. Notably, new results are in line with those obtained using the MRC5SV cells, rendering us confident that our findings are reliable overall. By contrast, we do not feel that including the WRNIP1 ATPase mutant cells is always essential, since our data clearly demonstrate that the loss of ATPase activity of WRNIP1 does not affect transcriptionassociated genome instability.

(2) The authors use the S9.6 antibody to conclude the loss of WRNIP1 causes more R-loops; however, it has been shown that this antibody detects dsRNA in addition to RNA-DNA hybrids. Accordingly, it cannot be ruled out that the increased S9.6 signal is due to increased dsRNA.

To eliminate contaminant-free RNA, particularly dsRNA, which could interfere with the capture of RNA-DNA hybrids by the S9.6 antibody (Hartono et al., 2018), and to determine R-loop levels more accurately, we treated cells with RNase III, following established protocols (Crossley et al., 2020). Under our experimental conditions, RNase III treatment significantly reduced the amount of dsRNA, nearly eliminating it, as evaluated using a specific antibody against dsRNA (see Suppl Fig 2 of the revised manuscript). To better appreciate the effect of the loss of WRNIP1 or its UBZ domain on Rloop accumulation and the amount of DNA damage, we have reproduced key data (see Figs 3B; 4D and E; 6B, D and E of the revised manuscript). Our analysis from immunofluorescence experiments, performed using a dsRNA ribonuclease, confirms higher R-loop accumulation in WRNIP1-deficient or UBZ WRNIP1 mutant cells compared to control cells (Fig. 3B). Additionally, proximity ligation assay (PLA) data are consistent with those previously presented and, in some cases, are more readily interpretable (see Figs 4D and E; 6B of the revised manuscript).

(3) Multiple pieces of data do not support the conclusions. For example, Figure 1D shows shWRNIP1 to reduce damage in Aph+DRB cells compared to MRC5SV cells with Aph+DRB. This result suggests that WRNIP1 actually increases DNA damage in stressed cells with transcription blocked. Another result is seen in Figure 4a, where the number of PLA spots (presumably TRCs) increases in the shWRNIP1WT cells with Aph+RNH1 compared to Aph alone. If R-loops are required for TRC accumulation, then the RNH1 should decrease the PLA foci. This result instead suggests that WRNIP leads to increased TRCs in stressed cells with R-loops cleared by RNH1.

Regarding Figure 1D, in MRC5SV cells, DRB does not significantly increase DNA damage upon Aph treatment. Therefore, it is not correct to conclude that WRNIP1 exacerbates DNA damage in stressed cells with transcription blocked.

Regarding Figure 4A, while the outcome may appear unexpected, and we do not provide data that explain the nature of this phenotype, a previous study demonstrated that overexpression of RNase H1 in mammalian cells may lead to a dose-dependent reduction of certain proteins of the repair pathway, leading to a significant accumulation of DNA damage (Shen et al., 2017). Accordingly, the observed increase in TRCs upon RNase H1 overexpression in wild-type cells may be attributed to the disruption of proteins that, by impairing the repair process, can potentially cause more fork stalling and, consequently, more conflicts. We have introduced a comment in the text.

(4) The data are mostly phenomenological and fail to yield mechanistic insight. For example, the authors state that "it remains unclear whether WRNIP1 is directly involved in the mechanisms of Rloop removal/resolution". Unfortunately, the data presented in this manuscript do not provide new insights into this unresolved question.

We agree with the reviewer that elucidating the mechanism by which WRNIP1 contributes to R-loop suppression would be of interest. Nevertheless, the findings presented here provide compelling evidence of a novel role for WRNIP1 in preventing R-loop accumulation. Investigating how WRNIP1 accomplishes this function will require significant effort, which we are committed to undertaking.

(5) The authors only show merged images making it impossible to visualize differences in PLA foci.

For a better visualization of nuclei signals in the PLA panels of Figs 4A, B, C, D and E; 6B, singlecolor images have been provided.

In addition to including the controls I mentioned in the public review, I recommend investigating the mechanism of how WRNIP1 prevents R-loop accumulation. If it is indeed related to its UBZ domain, then does that mean ubiquitination is an important step in R-loop removal? I believe elucidating this would be a novel and significant contribution. If it's not related to ubiquitination, then how does the UBZ domain regulate R-loops?

We agree with the reviewer that investigating the precise role of the UBZ domain of WRNIP1 in Rloop prevention would be of interest, and several experiments are required to adequately address this issue. However, as discussed, we hypothesize that the UBZ domain might contribute to directing WRNIP1 to DNA at TRC sites through RAD18.

I recommend using purified RNH1-dead-GFP to detect R-loops as opposed to the S9.6 antibody. The Cimprich lab has published this recently as a tool for detecting R-loops in fixed cells.

As explained in point (2), to eliminate contaminant-free RNA, particularly dsRNA, which could interfere with the capture of RNA-DNA hybrids by the S9.6 antibody (Hartono et al., 2018), and to determine R-loop levels more accurately, we used treatment with RNase III, following established protocols (Crossley et al., 2020). New experiments are reported in the revised version of the manuscript for R-loops in all cell lines (see Fig. 3B of the revised manuscript).

Additionally, colocalization by PLA of WRNIP1/R-loops, RAD51/R-loops, FANCD2/R-loops, and R-loop accumulation by anti-S9.6 antibody in cells depleted of FANCD2 are presented (see Figs. 4D and E; 6B and D of the revised manuscript).

Furthermore, we repeated the dot-blot experiment (see Fig. 2B of the revised manuscript). We treated the samples with RNase H to degrade RNA-DNA hybrids and hybridized the membrane with an antidsDNA antibody to quantify R-loop levels more accurately. Our analysis confirms that the S9.6 signal strongly accumulates in shWRNIP1 cells compared to shWRNIP1WT cells. Additionally, a graph illustrating the fold-change values of the S9.6/dsDNA signal relative to wild-type untreated cells is provided.

Importantly, overall, our findings suggest that treatment with RNase III does not substantially change the results obtained without it, but in some cases, such as in Fig. 4D, makes them are more readily interpretable. Specifically, the new protocol allows us to visualize a higher number of PLA foci, and Aph increased the spots per nucleus in shWRNIP1D37A cells compared to the previous experiment (see Fig. 4D of the revised manuscript).